# Hyperbolic Feature Augmentation via Distribution Estimation and Infinite Sampling on Manifolds

**Zhi Gao**[1], **Yuwei Wu**[1,2]*, **Yunde Jia**[2,1], **Mehrtash Harandi**[3]

[1] Beijing Lab of Intelligent Information Technology, School of Computer Science,
Beijing Institute of Technology, China
[2] Guangdong Lab of Machine Perception and Intelligent Computing,
Shenzhen MSU-BIT University, China
[3] Department of Electrical and Computer Systems Eng.,
Monash University, and Data61, Australia

{gaozhi_2017,wuyuwei,jiayunde}@bit.edu.cn, mehrtash.harandi@monash.edu

## Abstract

Learning in hyperbolic spaces has attracted growing attention recently, owing to their capabilities in capturing hierarchical structures of data. However, existing learning algorithms in the hyperbolic space tend to overfit when limited data is given. In this paper, we propose a hyperbolic feature augmentation method that generates diverse and discriminative features in the hyperbolic space to combat overfitting. We employ a wrapped hyperbolic normal distribution to model augmented features, and use a neural ordinary differential equation module that benefits from meta-learning to estimate the distribution. This is to reduce the bias of estimation caused by the scarcity of data. We also derive an upper bound of the augmentation loss, which enables us to train a hyperbolic model by using an infinite number of augmentations. Experiments on few-shot learning and continual learning tasks show that our method significantly improves the performance of hyperbolic algorithms in scarce data regimes.

## 1 Introduction

Several recent studies reveal that real-world data, such as images, videos, graphs, and sentences in natural language, usually endows a hierarchical structure [1]. In contrast to the Euclidean space with zero curvature, a hyperbolic space characterized by constant negative curvature can well capture such hierarchical structures, since the volume of the hyperbolic space grows exponentially with respect to the radius [2]. This property leads to rich representations and encourages successful developments of hyperbolic algorithms for various applications. Examples include image retrieval [3], action recognition [4], graph classification [5], and machine translation [6].

Although existing hyperbolic algorithms have achieved impressive performances, they often rely on the availability of sufficient data for training. In many practical applications, very limited data is given, as collecting and labeling data is costly. For example, in few-shot animal recognition [7, 8], animal images have hierarchical structures according to their species information. With few images (say less than five) in each class, training a model (*e.g.*, a ResNet combined with hyperbolic geometry [9, 3]) suffers from the undesirable overfitting problem. This motivates us to study combating overfitting with limited data in the hyperbolic space.

In this paper, we propose a hyperbolic feature augmentation (HFA) method that overcomes overfitting by generating class-identity-preserving features in the hyperbolic space. Compared with augmentation

---

*Corresponding author: Yuwei Wu

36th Conference on Neural Information Processing Systems (NeurIPS 2022).

in the original space of data (*e.g.*, image space), augmentation in the feature space requires less memory footprint [10, 11]. Moreover, feature augmentation is capable of providing useful intra-class variants by capturing diverse and discriminative augmentation directions, enabling the model to learn the invariance of each class [12, 13]. Therefore, feature augmentation offers a feasible way to overcome overfitting in the hyperbolic space.

Performing feature augmentation in the hyperbolic space involves two challenges. (1) The augmentation directions should be discriminative and reflect the variations of intra-class in the curved hyperbolic space rather than a flat space. The scarce data and complicated hierarchical structures make producing suitable augmentation directions non-trivial. (2) Augmentation in the hyperbolic space is computationally expensive (as compared to the Euclidean space), as the process of augmentation requires complex hyperbolic operations (*e.g.*, parallel transport and exponential map) to preserve the non-Euclidean geometry of augmented features.

We introduce a meta-neural-ODE distribution estimation scheme to solve the first challenge. We employ a wrapped normal distribution for augmented features, where the augmentation directions are sampled in the tangent bundle and projected to the hyperbolic space. We estimate the distribution by solving a continuous optimization process with neural ordinary differential equation (ODE), and utilize meta-learning to acquire prior knowledge in a gradient flow network for the estimation. The continuous optimization with prior knowledge leads to more reliable distribution for scarce data and complicated structures. We derive an upper bound of the augmentation loss to solve the second challenge, which enables us to train a hyperbolic model using an infinite number of sampled features. As a result, our method brings enhanced diversity of data and does not need to perform data augmentation explicitly, making an efficient augmentation algorithm without complex hyperbolic operations. We evaluate the proposed HFA method on few-shot learning and continual learning tasks, and results show that the proposed method significantly improves the performance of hyperbolic algorithms when limited data is given. The code of HFA is available at `https://github.com/ZhiGaomcislab/Hyperbolic_Feature_Augmentation`.

In summary, our contributions are threefold. **(1)** To the best of our knowledge, we are the first to perform feature augmentation in the hyperbolic space. The proposed HFA method is capable of generating diverse and discriminative hyperbolic features, addressing the overfitting problem. **(2)** We introduce a meta-neural-ODE distribution estimation scheme, whose continuous optimization process with prior knowledge leads to precise approximation of the real distribution in scarce data regimes. **(3)** We derive an upper bound of the augmentation loss to train a hyperbolic model using infinite augmentation, which creates an efficient augmentation algorithm without explicit augmentation and hyperbolic operations.

## 2   Related Work

**Data augmentation.** Data augmentation techniques, widely used to improve generalization, can be divided into two categories: **1.** augmentation in the original space of data (*e.g.*, image space) and **2.** augmentation in the feature space. Hand designed transformations (*e.g.*, masking and cropping) are popular examples from the first category [14, 15]. Automatic augmentation is a promising technique to automatically choose hand-crafted transformations [16, 17]. To produce more intra-class variants, some methods train a generative model for augmentation in the original space [18, 19]. One drawback of generating augmentation in the original space is the computational load of the algorithms. In contrast, feature augmentation not only brings rich intra-class variants but also involves less computational complexity and memory footprint. The benefits of feature augmentation are demonstrated for many applications, including but not limited to contrastive learning [20], domain adaptation [21], few-shot learning [22], long-tailed recognition [13], and instance segmentation [11]. Feature augmentation is usually achieved by manifold mixup [23, 24], sampling from captured distributions [25, 12, 10, 11], or utilizing a GAN model [26, 27]. Different from existing feature augmentation, we focus on augmentation in the hyperbolic space.In doing so, we introduce a meta-neural-ODE distribution estimation scheme, instead of directly computing distribution from given data. In this case, our method can better adapt to complex distributions in scarce data regimes.

**Hyperbolic geometry.** Learning in the hyperbolic space has become an alternative to the Euclidean space, since natural data usually exhibits hierarchical structures. Existing efforts of exploiting the hyperbolic space can be roughly divided into two categories: learning hyperbolic embeddings and

designing hyperbolic neural networks. Methods of the first category add several hyperbolic operations on the top of a conventional neural network to obtain discriminative hyperbolic embeddings. This scheme has been used in many applications, such as image retrieval [3], few-shot learning [28], metric learning [29], and image segmentation [30]. The second category focuses on extending the entire architecture of a neural network to the hyperbolic space, such as hyperbolic convolutional network [6], hyperbolic graph network [31, 5], hyperbolic attention network [32], and hyperbolic variational autoencoder [33, 34]. These methods mainly focus on the model level of hyperbolic algorithms (*i.e.*, designing/improving hyperbolic models). In contrast, our method studies the data level of hyperbolic algorithms and efficiently improves the generalization of models via data augmentation.

## 3   Preliminaries

A hyperbolic space is a smooth Riemannian manifold with constant negative curvature [35]. Hyperbolic space has five isometric models, and we use the Poincaré ball model [36] to work with. A Poincaré ball is defined as $\mathcal{M}^{d,c} = \{\boldsymbol{x} \in \mathbb{R}^d, -c\|\boldsymbol{x}\| < 1\}$, where $d$ is the dimension of vectors in the space, $\|\cdot\|$ is the Euclidean norm, and the curvature $c < 0$ represents the deviation of $\mathcal{M}^{d,c}$ from a flat space. For $\boldsymbol{u} \in \mathcal{M}^{d,c}$, its tangent space, denoted by $T_{\boldsymbol{u}}\mathcal{M}^{d,c}$, contains all tangent vectors to $\mathcal{M}^{d,c}$ at $\boldsymbol{u}$. We will make use of the following operations in our work;

**Addition.** For $\boldsymbol{x}, \boldsymbol{y} \in \mathcal{M}^{d,c}$, their addition is

$$\boldsymbol{x} \oplus_c \boldsymbol{y} = \frac{(1 - 2c\langle \boldsymbol{x}, \boldsymbol{y}\rangle_2 - c\|\boldsymbol{y}\|^2)\boldsymbol{x} + (1 + c\|\boldsymbol{x}\|^2)\boldsymbol{y}}{1 - 2c\langle \boldsymbol{x}, \boldsymbol{y}\rangle_2 + c^2\|\boldsymbol{x}\|^2\|\boldsymbol{y}\|^2}. \tag{1}$$

**Distance measure.** For two vectors $\boldsymbol{x}, \boldsymbol{y} \in \mathcal{M}^{d,c}$, their distance is

$$d^c(\boldsymbol{x}, \boldsymbol{y}) = \frac{1}{\sqrt{|c|}}\cosh^{-1}\left(1 - 2c\frac{\|\boldsymbol{x} - \boldsymbol{y}\|^2}{(1 + c\|\boldsymbol{x}\|^2)(1 + c\|\boldsymbol{y}\|^2)}\right). \tag{2}$$

**Exponential map.** The exponential map $\text{expm}_{\boldsymbol{x}}^c(\boldsymbol{s})$ projects a vector $\boldsymbol{s}$ from the tangent space $T_{\boldsymbol{x}}\mathcal{M}^{d,c}$ to the manifold $\mathcal{M}^{d,c}$,

$$\text{expm}_{\boldsymbol{x}}^c(\boldsymbol{s}) = \boldsymbol{x} \oplus_c \left(\tanh(\sqrt{|c|}\frac{\lambda_{\boldsymbol{x}}^c\|\boldsymbol{s}\|}{2})\frac{\boldsymbol{s}}{\sqrt{|c|}\|\boldsymbol{s}\|}\right), \tag{3}$$

where $\lambda_{\boldsymbol{x}}^c = 2/(1 + c\|\boldsymbol{x}\|^2)$ is the conformal factor. The exponential map in the tangent space $T_{\boldsymbol{0}}\mathcal{M}^{d,c}$ at the origin is simplified to $\text{expm}_{\boldsymbol{0}}^c(\boldsymbol{s}) = \tanh(\sqrt{|c|}\|\boldsymbol{s}\|)\frac{\boldsymbol{s}}{\sqrt{|c|}\|\boldsymbol{s}\|}$.

**Logarithmic map.** The logarithmic map $\text{logm}_{\boldsymbol{x}}^c(\boldsymbol{y})$ maps a vector $\boldsymbol{y}$ from the manifold to the tangent space $T_{\boldsymbol{x}}\mathcal{M}^{d,c}$,

$$\text{logm}_{\boldsymbol{x}}^c(\boldsymbol{y}) = \frac{2}{\sqrt{|c|}\lambda_{\boldsymbol{x}}^c}\tanh^{-1}(\sqrt{|c|}\| - \boldsymbol{x} \oplus_c \boldsymbol{y}\|)\frac{-\boldsymbol{x} \oplus_c \boldsymbol{y}}{\| - \boldsymbol{x} \oplus_c \boldsymbol{y}\|}. \tag{4}$$

The exponential map $\text{logm}_{\boldsymbol{0}}^c$ in $T_{\boldsymbol{0}}\mathcal{M}^{d,c}$ is simplified to $\text{logm}_{\boldsymbol{0}}^c(\boldsymbol{y}) = \tanh^{-1}(\sqrt{|c|}\|\boldsymbol{y}\|)\frac{\boldsymbol{y}}{\sqrt{|c|}\|\boldsymbol{y}\|}$.

**Parallel transport** The parallel transport operation $\text{PT}_{\boldsymbol{x} \to \boldsymbol{y}}^c(\boldsymbol{s}) : T_{\boldsymbol{x}}\mathcal{M}^{d,c} \to T_{\boldsymbol{y}}\mathcal{M}^{d,c}$ moves a vector from the tangent space $T_{\boldsymbol{x}}\mathcal{M}^{d,c}$ to another one $T_{\boldsymbol{y}}\mathcal{M}^{d,c}$ along the geodesic,

$$\text{PT}_{\boldsymbol{x} \to \boldsymbol{y}}^c(\boldsymbol{s}) = \frac{\lambda_{\boldsymbol{x}}^c}{\lambda_{\boldsymbol{y}}^c}\text{gyr}[\boldsymbol{y}, -\boldsymbol{x}]\boldsymbol{s}, \tag{5}$$

where $\text{gyr}[\boldsymbol{y}, -\boldsymbol{x}]\boldsymbol{s} = -(\boldsymbol{x} \oplus_c \boldsymbol{y}) \oplus_c (\boldsymbol{x} \oplus_c (\boldsymbol{y} \oplus_c \boldsymbol{s}))$. This operation in $T_{\boldsymbol{0}}\mathcal{M}^{d,c}$ is $\text{PT}_{\boldsymbol{0} \to \boldsymbol{y}}^c(\boldsymbol{s}) = \frac{2}{\lambda_{\boldsymbol{y}}^c}\boldsymbol{s}$.

## 4   Method

### 4.1   Formulation

In this paper, we propose the hyperbolic feature augmentation (HFA) method. We employ a wrapped normal distribution $\mathcal{P}(c, \boldsymbol{p}, \boldsymbol{\mu}, \boldsymbol{\Sigma})$ to model features in scarce data regimes, and meta-learn a neural

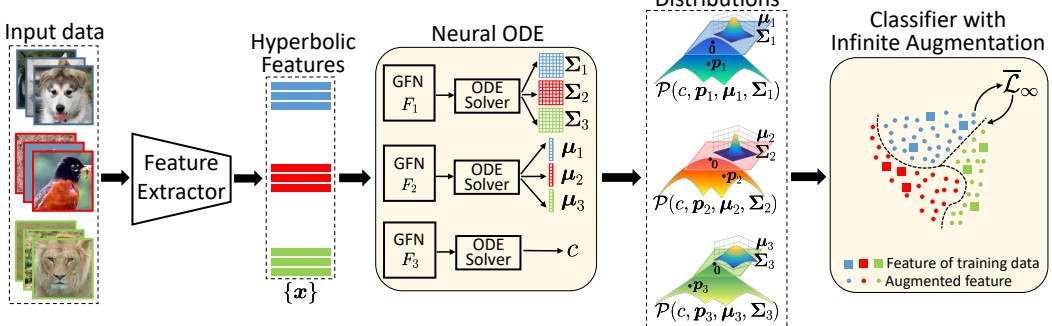

Figure 1: A conceptual diagram of HFA. GFN denotes the proposed gradient flow network.

ODE to estimate a distribution for each class. Then, we derive an upper bound $\overline{\mathcal{L}}_\infty$ of the loss function with infinite augmentation to train the classifier. An illustration of main components of HFA is provided in Figure 1.

## 4.2 Wrapped Normal Distribution in Scarce Data Regime

The wrapped normal distribution has been successfully used in various studies to model distributions in Riemannian manifolds [37, 38, 39]. The underlying idea is to model a Riemannian distribution as a projection of a normal distribution on a tangent space to the manifold. Here, we introduce it to scarce data regimes. The density of a wrapped normal distribution is given by

$$\mathcal{P}(\boldsymbol{z}|c,\boldsymbol{p},\boldsymbol{\mu},\boldsymbol{\Sigma}) = \mathcal{N}(\lambda_{\boldsymbol{p}}^c \mathrm{logm}_{\boldsymbol{p}}^c(\boldsymbol{z})|\boldsymbol{\mu},\boldsymbol{\Sigma}) \left( \frac{\sqrt{|c|}d^c(\boldsymbol{p},\boldsymbol{z})}{\sinh(\sqrt{|c|}d^c(\boldsymbol{p},\boldsymbol{z}))} \right), \qquad (6)$$

where $c$ is the curvature of the underlying space, $\boldsymbol{p} \in \mathcal{M}^{d,c}$ is the prototype of given data, and $\boldsymbol{\mu}$ and $\boldsymbol{\Sigma} \in \mathbb{R}^{d \times d}$ are the mean and the covariance matrix of augmentation directions, respectively. Since few data is given in scarce data regimes, the prototype $\boldsymbol{p}$ may be biased.

Generating a feature from $\mathcal{P}(c,\boldsymbol{p},\boldsymbol{\mu},\boldsymbol{\Sigma})$ includes three steps. (1) Sample a vector $\widehat{\boldsymbol{v}}$ from the normal distribution $\mathcal{N}(\boldsymbol{\mu},\boldsymbol{\Sigma})$ in the tangent space $T_{\boldsymbol{0}}\mathcal{M}^{c,d}$. (2) Transport $\widehat{\boldsymbol{v}}$ from $T_{\boldsymbol{0}}\mathcal{M}^{c,d}$ to $\boldsymbol{v}$ in the tangent space $T_{\boldsymbol{p}}\mathcal{M}^{c,d}$, $\boldsymbol{v} = \mathrm{PT}_{\boldsymbol{0}\to\boldsymbol{p}}^c(\widehat{\boldsymbol{v}})$. (3) Map $\boldsymbol{v}$ to the manifold by $\boldsymbol{z} = \mathrm{expm}_{\boldsymbol{p}}^c(\boldsymbol{v})$. In the step (1), sampling $\widehat{\boldsymbol{v}}$ from $\mathcal{N}(\boldsymbol{\mu},\boldsymbol{\Sigma})$ is not a differentiable process, and we can not back propagate gradients to $\boldsymbol{\mu}$ and $\boldsymbol{\Sigma}$. We use a reparameterization trick to solve this issue. We sample a vector $\boldsymbol{\epsilon}$ from the standard normal distribution $\mathcal{N}(\boldsymbol{0},\boldsymbol{I})$ and transform $\boldsymbol{\epsilon}$ to $\widehat{\boldsymbol{v}}$ by $\widehat{\boldsymbol{v}} = \boldsymbol{\mu} + \boldsymbol{L}\boldsymbol{\epsilon}$, where $\boldsymbol{L}\boldsymbol{L}^\top = \boldsymbol{\Sigma}$. The sampling process is summarized in Algorithm 1.

---

**Algorithm 1** Sampling process in HFA.

**Input:** A wrapped normal distribution $\mathcal{P}(c,\boldsymbol{p},\boldsymbol{\mu},\boldsymbol{\Sigma})$, where $\boldsymbol{\Sigma}$ is decomposed by $\boldsymbol{L}\boldsymbol{L}^\top = \boldsymbol{\Sigma}$.
**Output:** $\boldsymbol{z} \in \mathcal{M}^{d,c}$ sampled from $\mathcal{P}$
1: Sample $\boldsymbol{\epsilon}$ from a standard normal distribution $\mathcal{N}(\boldsymbol{0},\boldsymbol{I})$.
2: Use the reparameterization trick to obtain $\widehat{\boldsymbol{v}} = \boldsymbol{\mu} + \boldsymbol{L}\boldsymbol{\epsilon}$.
3: Transport $\widehat{\boldsymbol{v}}$ from $T_{\boldsymbol{0}}\mathcal{M}^{c,d}$ to $T_{\boldsymbol{p}}\mathcal{M}^{c,d}$ by $\boldsymbol{v} = \mathrm{PT}_{\boldsymbol{0}\to\boldsymbol{p}}^c(\widehat{\boldsymbol{v}})$.
4: Map $\boldsymbol{v}$ to the manifold by $\boldsymbol{z} = \mathrm{expm}_{\boldsymbol{p}}^c(\boldsymbol{v})$.

---

We apply one wrapped normal distribution to each class. For the distribution $\mathcal{P}(c,\boldsymbol{p}_j,\boldsymbol{\mu}_j,\boldsymbol{\Sigma}_j)$ of the $j$-th class, its has its own prototype $\boldsymbol{p}_j$, mean $\boldsymbol{\mu}_j$, and covariance matrix $\boldsymbol{\Sigma}_j$. All classes share a common curvature $c$.

## 4.3 Distribution Estimation by Neural ODE

**Neural ODE.** For the wrap distribution $\mathcal{P}(c,\boldsymbol{p}_j,\boldsymbol{\mu}_j,\boldsymbol{\Sigma}_j)$, the prototype $\boldsymbol{p}_j$ can be computed by averaging given samples (*e.g.*, the Einstein midpoint [40]). We use a neural ODE to estimate $c$, $\boldsymbol{\mu}_j$, and $\boldsymbol{L}_j$, by viewing an iteration process as an Euler discretization of an ODE [41, 42, 43]. $\boldsymbol{\Sigma}_j$ is computed by $\boldsymbol{\Sigma}_j = \boldsymbol{L}_j \boldsymbol{L}_j^\top$. Concretely, we denote the distribution parameter as $\xi$ (it can be $c$, $\boldsymbol{\mu}_j$, or

$\boldsymbol{L}_j$) and the estimation of $\xi$ can be cast as an iterative optimization process using given data (*e.g.*, maximizing the likelihood), $\xi^{t+1} = \xi^t + \nabla\xi^t$, where $\nabla\xi^t$ is the gradient at step $t$. We regard the iterative optimization process as a continues-time process of an ODE, and propose a gradient flow network $F$ to estimate the gradient flow $\frac{d\xi^t}{dt}$ at time $t$, $\nabla\xi^t = \frac{d\xi^t}{dt} = F(\xi^t, t)$. Thus, given the initial value $\xi^0$ and $F$, the distribution parameter is obtained by solving the ODE at the last time $T$ with an integral term: $\xi^T = \xi^0 + \int_{t=0}^{T} F(\xi^t, t)dt$.

Here, we utilize the Runge-Kutta method [44] denoted by ODESolve($\cdot$) to solve the integral term,

$$\xi^T = \text{ODESolve}(\xi^0, F, T). \tag{7}$$

In implementation, we use three gradient flow networks $F_1$, $F_2$, and $F_3$ to produce the gradient flow for $c$, $\boldsymbol{\mu}_j$, and $\boldsymbol{L}_j$, respectively. $\xi^0$ for $\boldsymbol{\mu}_j$ and $\boldsymbol{L}_j$ is obtained by computing the mean of given data and subtracting the mean from given data, and $\xi^0$ for $c$ is set as a fixed value.

**Gradient flow network.** This network takes the distribution parameter $\xi^t$ and features of a set of samples as inputs, and generates the gradient $\frac{d\xi^t}{dt}$ via three layers, as shown in Figure 2. We first build an estimation representation $\boldsymbol{e}_j$ for each class, where $\xi^t$ and the mean $\overline{\boldsymbol{x}}_j$ of features belonging to the $j$-th class are concatenated as $\boldsymbol{e}_j = [\xi^t, \overline{\boldsymbol{x}}_j]$.

We use a fully-connected layer $f_e$ to embed $\boldsymbol{e}_j$ as

$$\boldsymbol{e}'_j = f_e(\boldsymbol{e}_j). \tag{8}$$

Then, we use a self-attention layer $f_t$ to perform interaction among different classes,

$$\boldsymbol{e}''_j = f_t\left(\{\boldsymbol{e}'_j\}_{j=1}^n\right). \tag{9}$$

where the query, key, and value of the self-attention layer are all $\{\boldsymbol{e}'_j\}_{j=1}^n$ of classes, with $n$ classes in total. Finally, we use a fully-connected layer $f_o$ to create the gradient,

$$\frac{d\xi^t}{dt} = f_o(\{\boldsymbol{e}''_j\}). \tag{10}$$

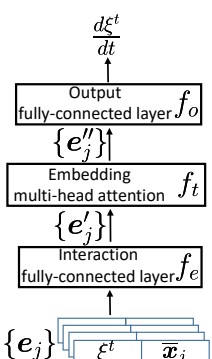

Figure 2: Architecture of the gradient flow network.

Note that, we employ a common curvature $c$ for all classes using $F_1$, instead of producing class-specific curvatures and unifying them. We send all data of the $n$ classes together to the gradient flow network $F_1$, concatenate the representation of all classes $\{\boldsymbol{e}''_j\}_{j=1}^n$, and use the output fully-connected layer $f_o$ to produce the gradient flow for the common curvature $c$.

## 4.4 Learning with Infinite Data Augmentation

We use a distance-based classifier in the hyperbolic space, and a feature is classified by assigning it to the closest weight of classes. We represent the weight of the $j$-th class as $\boldsymbol{w}_j \in \mathcal{M}^{d,c}$, and the classifier is denoted by $\boldsymbol{W} = [\boldsymbol{w}_1, \cdots, \boldsymbol{w}_n]$. The probability that a hyperbolic feature $\boldsymbol{x}$ belongs to the $j$-th class is

$$p(\widehat{y} = j|\boldsymbol{x}) = \frac{\exp\left(-d^c(\boldsymbol{x}, \boldsymbol{w}_j)\right)}{\sum_{j'=1}^n \exp\left(-d^c(\boldsymbol{x}, \boldsymbol{w}_{j'})\right)}, \tag{11}$$

where $\widehat{y}$ is the prediction of the classifier.

We sample features from the estimated distribution to train the classifier. We denote the augmented features for the $j$-th class by $\{\boldsymbol{z}_{j,i}\}_{i=1}^m$, where $\boldsymbol{z}_{j,i} \sim \mathcal{P}(c, \boldsymbol{p}_j, \boldsymbol{\mu}_j, \boldsymbol{\Sigma}_j)$ (a total of $m$ samples are generated per class). The classifier is trained by minimizing the following cross-entropy loss function,

$$\mathcal{L}(\boldsymbol{W}) = \frac{1}{n}\sum_{j=1}^n \frac{1}{m}\sum_{i=1}^m -\log\frac{\exp\left(-d^c(\boldsymbol{z}_{j,i}, \boldsymbol{w}_j)\right)}{\sum_{j'=1}^n \exp\left(-d^c(\boldsymbol{z}_{j,i}, \boldsymbol{w}_{j'})\right)}. \tag{12}$$

A large value of $m$ brings more diversity to the model, promising the model to learn the invariance of each class. Considering $m$ grows to infinity, the loss function is defined by the expectation of the cross-entropy,

$$\mathcal{L}_\infty(\boldsymbol{W}) = \frac{1}{n}\sum_{j=1}^n \mathbb{E}_i\left[-\log\frac{\exp\left(-d^c(\boldsymbol{z}_{j,i}, \boldsymbol{w}_j)\right)}{\sum_{j'=1}^n \exp\left(-d^c(\boldsymbol{z}_{j,i}, \boldsymbol{w}_{j'})\right)}\right]. \tag{13}$$

Computing Eq. (13) is difficult and the computational load of if MCMC methods are used for a large value of $m$. Alternatively, we derive an upper bound $\overline{\mathcal{L}}_\infty(\boldsymbol{W})$ of $\mathcal{L}_\infty(\boldsymbol{W})$, which is easy to compute, as shown in Proposition 1. In this case, we train the classifier by minimizing the upper bound $\overline{\mathcal{L}}_\infty(\boldsymbol{W})$, rather than the original loss function $\mathcal{L}_\infty(\boldsymbol{W})$.

**Proposition 1.** *Given infinite data $\boldsymbol{z}_{j,i} \sim \mathcal{P}(c, \boldsymbol{p}_j, \boldsymbol{\mu}_j, \boldsymbol{\Sigma}_j)$, an upper bound of the loss function $\mathcal{L}_\infty$ in Eq.* (13) *is given by*

$$
\overline{\mathcal{L}}_\infty(\boldsymbol{W}) = \frac{1}{n}\sum\nolimits_{j=1}^{n} -\log \frac{\exp\left(\boldsymbol{w}_j^\top(\widehat{\boldsymbol{p}}_j + \boldsymbol{\mu}_i)\right)}{\sum_{j'=1}^{n}\exp\left(\boldsymbol{w}_{j'}^\top(\widehat{\boldsymbol{p}}_j + \boldsymbol{\mu}_i) + (\boldsymbol{w}_{j'}^\top\boldsymbol{w}_{j'} - \boldsymbol{w}_j^\top\boldsymbol{w}_j) + \frac{1}{2}(\boldsymbol{w}_{j'} - \boldsymbol{w}_j)^\top\boldsymbol{\Sigma}_j(\boldsymbol{w}_{j'} - \boldsymbol{w}_j)\right)},
\tag{14}
$$

*where $\boldsymbol{p}_j = \mathrm{expm}_{\boldsymbol{0}}^c(\widehat{\boldsymbol{p}}_j)$, and $\widehat{\boldsymbol{p}}_j$ can be directly computed in the tangent space at the origin.*

*Proof.* We rewrite the loss function of the infinite data augmentation in Eq. (13) as

$$
\mathcal{L}_\infty(\boldsymbol{W}) = \frac{1}{n}\sum_{j=1}^{n}\mathbb{E}_i\left[-\log\left(\frac{\exp\left(-d^c(\boldsymbol{z}_{j,i}, \boldsymbol{w}_j)\right)}{\sum_{j'=1}^{n}\exp\left(-d^c(\boldsymbol{z}_{j,i}, \boldsymbol{w}_{j'})\right)}\right)\right]
$$

$$
= \frac{1}{n}\sum_{j=1}^{n}\mathbb{E}_i\left[\log\left(\sum_{j'=1}^{n}\exp\left(d^c(\boldsymbol{z}_{j,i}, \boldsymbol{w}_j) - d^c(\boldsymbol{z}_{j,i}, \boldsymbol{w}_{j'})\right)\right)\right]
$$

$$
\leq \frac{1}{n}\sum_{j=1}^{n}\log\left(\sum_{j'=1}^{n}\mathbb{E}_i\left[\exp\left(d^c(\boldsymbol{z}_{j,i}, \boldsymbol{w}_j) - d^c(\boldsymbol{z}_{j,i}, \boldsymbol{w}_{j'})\right)\right]\right)
\tag{15a}
$$

$$
\leq \frac{1}{n}\sum_{j=1}^{n}\log\left(\sum_{j'=1}^{n}\mathbb{E}_i\left[\exp\left(\|\boldsymbol{z}_{j,i} - \boldsymbol{w}_j\|^2 - \|\boldsymbol{z}_{j,i} - \boldsymbol{w}_{j'}\|^2\right)\right]\right)
\tag{15b}
$$

$$
= \frac{1}{n}\sum_{j=1}^{n}\log\left(\sum_{j'=1}^{n}\mathbb{E}_i\left[\exp\left(2(\boldsymbol{w}_{j'} - \boldsymbol{w}_j)^\top\boldsymbol{z}_{j,i} + (\boldsymbol{w}_{j'}^\top\boldsymbol{w}_{j'} - \boldsymbol{w}_j^\top\boldsymbol{w}_j)\right)\right]\right)
$$

$$
\leq \frac{1}{n}\sum_{j=1}^{n}\log\left(\sum_{j'=1}^{n}\mathbb{E}_i\left[\exp\left((\boldsymbol{w}_{j'} - \boldsymbol{w}_j)^\top(\widehat{\boldsymbol{p}}_j + \widehat{\boldsymbol{v}}_{j,i}) + (\boldsymbol{w}_{j'}^\top\boldsymbol{w}_{j'} - \boldsymbol{w}_j^\top\boldsymbol{w}_j)\right)\right]\right)
\tag{15c}
$$

$$
= \frac{1}{n}\sum_{j=1}^{n}\log\left(\sum_{j'=1}^{n}\exp\left((\boldsymbol{w}_{j'} - \boldsymbol{w}_j)^\top(\widehat{\boldsymbol{p}}_j + \boldsymbol{\mu}_j) + (\boldsymbol{w}_{j'}^\top\boldsymbol{w}_{j'} - \boldsymbol{w}_j^\top\boldsymbol{w}_j) + \frac{1}{2}(\boldsymbol{w}_{j'} - \boldsymbol{w}_j)^\top\boldsymbol{\Sigma}_j(\boldsymbol{w}_{j'} - \boldsymbol{w}_j)\right)\right)
\tag{15d}
$$

$$
= \frac{1}{n}\sum_{j=1}^{n} -\log\frac{\exp\left(\boldsymbol{w}_j(\widehat{\boldsymbol{p}}_j + \boldsymbol{\mu}_j)\right)}{\exp\left(\boldsymbol{w}_{j'}(\widehat{\boldsymbol{p}}_j + \boldsymbol{\mu}_j) + (\boldsymbol{w}_{j'}^\top\boldsymbol{w}_{j'} - \boldsymbol{w}_j^\top\boldsymbol{w}_j) + \frac{1}{2}(\boldsymbol{w}_{j'} - \boldsymbol{w}_j)^\top\boldsymbol{\Sigma}_j(\boldsymbol{w}_{j'} - \boldsymbol{w}_j)\right)}
$$

$$
= \overline{\mathcal{L}}_\infty(\boldsymbol{W}).
$$

Eq. (15a) follows from the Jensen's inequality $\mathbb{E}[\log(\boldsymbol{x})] \leq \log\mathbb{E}[\boldsymbol{x}]$. For Eq. (15b), we define a function $f = d^c(\boldsymbol{x}, \boldsymbol{y}) - \|\boldsymbol{x} - \boldsymbol{y}\|^2$ and compute the derivative of $f$. We find $f$ is an increasing function when $\|\boldsymbol{x} - \boldsymbol{y}\|^2 < \frac{\sqrt{1+|c|} - \sqrt{|c|}}{\sqrt{|c|}}$. In this case, if we have $\|\boldsymbol{z}_{j,i} - \boldsymbol{w}_j\|^2 < \|\boldsymbol{z}_{j,i} - \boldsymbol{w}_{j'}\|^2 < \frac{\sqrt{1+|c|} - \sqrt{|c|}}{\sqrt{|c|}}$, we can derive $d^c(\boldsymbol{z}_{j,i}, \boldsymbol{w}_j) - \|\boldsymbol{z}_{j,i}, \boldsymbol{w}_j\|^2 < d^c(\boldsymbol{z}_{j,i}, \boldsymbol{w}_{j'}) - \|\boldsymbol{z}_{j,i}, \boldsymbol{w}_{j'}\|^2$ and $\|\boldsymbol{z}_{j,i}, \boldsymbol{w}_j\|^2 - \|\boldsymbol{z}_{j,i}, \boldsymbol{w}_{j'}\|^2 > d^c(\boldsymbol{z}_{j,i}, \boldsymbol{w}_j) - d^c(\boldsymbol{z}_{j,i}, \boldsymbol{w}_{j'})$. Eq. (15c) is obtained by replacing $\boldsymbol{z}_{j,i}$ with the augmentation process using the parallel transport and exponential map, *i.e.*, $\boldsymbol{z}_{j,i} = \mathrm{expm}_{\boldsymbol{p}_j}^c(\mathrm{PT}_{\boldsymbol{0}\to\boldsymbol{p}_j}^c(\widehat{\boldsymbol{v}}_{j,i}))$, and adding scale constraints on $c$, $\|\widehat{\boldsymbol{v}}_{j,i}\|$, and $\|\widehat{\boldsymbol{p}}_j\|$. Eq. (15d) is obtained by using the moment-generating function: $\mathbb{E}[\exp(\boldsymbol{x})] = \exp(\boldsymbol{\mu} + \frac{1}{2}\boldsymbol{\Sigma})$, $\boldsymbol{x} \sim \mathcal{N}(\boldsymbol{\mu}, \boldsymbol{\Sigma})$. In our derivation, $(\boldsymbol{w}_{j'} - \boldsymbol{w}_j)^\top(\widehat{\boldsymbol{p}}_j + \widehat{\boldsymbol{v}}_{j,i}) + (\boldsymbol{w}_{j'}^\top\boldsymbol{w}_{j'} - \boldsymbol{w}_j^\top\boldsymbol{w}_j)$ is a Gaussian vector. $\square$

Detailed derivation and implementation can be found in the supplementary materials. From this derivation, we replace a non-Euclidean loss function with a Euclidean upper bound by adding suitable constraints. The upper bound is easy to compute without explicit augmentation and time-consumption hyperbolic operations, and our experimental results show that it achieves good performance (see

Section 5). This provides an interesting research direction and feasible way for Riemannian algorithms whose loss functions are difficult or high-cost to compute.

The classifier $\boldsymbol{W}$ is updated by the Riemannian gradient descent algorithm [45] in the process of minimizing $\overline{\mathcal{L}}_\infty(\boldsymbol{W})$. After several iterations, we obtain the updated classifier $\boldsymbol{W}^* = [\boldsymbol{w}_1^*, \cdots, \boldsymbol{w}_n^*]$, and test data is classified by Eq. (11).

## 4.5 Training

The goal of this work is to learn the gradient flow networks $F_1$, $F_2$, and $F_3$. In doing so, we train them in a meta-learning framework that simulates the practical setting by using some base data $\mathcal{D}$. In the training stage, we randomly select few data from $\mathcal{D}$ as the training data $\mathcal{D}_t$, and the rest is used as the validation data $\mathcal{D}_v$. $F_1$, $F_2$, and $F_3$ are updated via a bi-level optimization manner. In the inner-loop, we estimate the distribution of $\mathcal{D}_t$ by using $F_1$, $F_2$, and $F_3$, iteratively train the classifier $\boldsymbol{W}$ by minimizing the loss function $\overline{\mathcal{L}}_\infty$, and obtain the updated classifier $\boldsymbol{W}^* = [\boldsymbol{w}_1^*, \cdots, \boldsymbol{w}_n^*]$. We update $F_1$, $F_2$, and $F_3$ in the outer-loop by minimizing the following objective that is the loss of $\boldsymbol{W}^*$ on the validation data $\mathcal{D}_v$,

$$
\min_{F_1, F_2, F_3} \mathcal{L}(\boldsymbol{W}^*) = \mathbb{E}_{\boldsymbol{x} \sim \mathcal{D}_v} \left[ -\sum_{j=1}^n \mathbb{1}_{y=j} \log \frac{\exp\left(-d^c(\boldsymbol{x}, \boldsymbol{w}_j^*)\right)}{\sum_{j'=1}^n \exp\left(-d^c(\boldsymbol{x}, \boldsymbol{w}_{j'}^*)\right)} \right], \tag{16}
$$
$$
\text{s.t. } \boldsymbol{W}^* = \arg\min_{\boldsymbol{W}} \overline{\mathcal{L}}_\infty(\boldsymbol{W})
$$

where $\mathbb{1}_{y=j}$ is the indicator function, meaning that if the label $y$ of $\boldsymbol{x}$ is equal to $j$, $\mathbb{1}_{y=j} = 1$, and $0$ otherwise. The pseudo code of training is summarized in Algorithm 2.

In the inference stage, given a task with limited data, our method first estimates the distribution for each class by using the learned $F_1$, $F_2$, and $F_3$, and then trains the model by using $\overline{\mathcal{L}}_\infty$.

## 5 Experiments

We evaluate the proposed method HFA on few-shot learning and continual learning tasks. In few-shot learning, data augmentation is a natural idea to overcome the deficiency of training data [25, 26]. In continual learning, reply-based methods have shown effectiveness in continual learning by storing data [46, 47], yet they cannot store much data due to the memory limitation. We use HFA to perform augmentation for stored

---

**Algorithm 2** Training process of HFA.

**Input:** Base data $\mathcal{D}$.
**Output:** Updated gradient flow networks $F_1$, $F_2$, $F_3$.
1: **while** Not converged **do**
2:     Randomly select few training data $\mathcal{D}_t$ from $\mathcal{D}$, and the rest of $\mathcal{D}$ is used as validation data $\mathcal{D}_v$.
3:     Estimate the distribution for each class in $\mathcal{D}_t$ by using $F_1$, $F_2$, and $F_3$ via Eq. (7).
4:     **while** Not converged **do**
5:         Train the classifier $\boldsymbol{W}$ using the estimated distribution via the loss function in Eq. (14).
6:     **end while**
7:     Compute the loss of the updated classifier $\boldsymbol{W}^*$ using the validation data $\mathcal{D}_v$, and update $F_1$, $F_2$, and $F_3$ via minimizing Eq. (16).
8: **end while**

---

limited data to solve this issue. We use common backbone networks with exponential map as the feature extractor in the two tasks. More experiments (*e.g.*, evaluation on graph data, visualization, and ablation) and experimental details can be found in the supplementary materials.

### 5.1 Few-shot Learning

We conducted experiments on four few-shot learning datasets: mini-ImageNet [48], tiered-ImageNet [49], CUB [7], and CIFAR-FS [50] datasets. The four datasets have hyperbolic structures, and are commonly used to evaluate hyperbolic algorithms [3, 9, 29, 28]. We use ResNet12 and ResNet18 as the backbone networks, and perform augmentation for the support data.

### 5.1.1 Main Results

We compare HFA with augmentation-based few-shot learning methods [22, 51, 25, 26, 52, 53, 54, 55, 27] on the mini-ImageNet, CUB, and CIFAR-FS datasets. These methods all augment data in Euclidean space, while HFA performs data augmentation in the hyperbolic space. Our method brings improvements on them, as shown in Table 1. In particular, VFSL [54] and V1-Net [25] study data

Table 1: Accuracy (%) comparisons with Euclidean augmentation-based few-shot learning methods on the mini-ImageNet, CUB, and CIFAR-FS datasets.

| Method | Backbone | mini-ImageNet | | CUB | | CIFAR-FS | |
|---|---|---|---|---|---|---|---|
| | | 1-shot | 5-shot | 1-shot | 5-shot | 1-shot | 5-shot |
| VFSL [54] | ResNet12 | $61.23 \pm 0.26$ | $77.69 \pm 0.17$ | - | - | - | - |
| DTN [55] | ResNet12 | $63.45 \pm 0.86$ | $77.91 \pm 0.62$ | 72.00 | 85.10 | 71.50 | 82.80 |
| **Ours** | ResNet12 | $\mathbf{66.87 \pm 0.44}$ | $\mathbf{82.08 \pm 0.31}$ | $\mathbf{76.75 \pm 0.43}$ | $\mathbf{89.19 \pm 0.26}$ | $\mathbf{72.52 \pm 0.46}$ | $\mathbf{85.33 \pm 0.33}$ |
| Delta-encoder [52] | VGG-16 | 59.9 | 69.7 | $69.80 \pm 0.46$ | $82.60 \pm 0.35$ | 66.70 | 79.80 |
| SalNet [53] | ResNet101 | $62.22 \pm 0.87$ | $77.95 \pm 0.65$ | - | - | - | - |
| Dual TriNet [22] | ResNet18 | $58.12 \pm 1.37$ | $76.92 \pm 0.69$ | 69.61 | 84.10 | $63.41 \pm 0.64$ | $78.43 \pm 0.64$ |
| IDeMe-Net [51] | ResNet18 | $59.14 \pm 0.86$ | $74.63 \pm 0.74$ | - | - | - | - |
| V1-Net [25] | ResNet18 | 61.05 | 78.60 | 74.76 | 86.84 | - | - |
| AFHN [26] | ResNet18 | $62.38 \pm 0.72$ | $78.16 \pm 0.56$ | $70.53 \pm 1.01$ | $83.95 \pm 0.63$ | $68.32 \pm 0.93$ | $81.45 \pm 0.87$ |
| TFH [27] | ResNet18 | $65.07 \pm 0.82$ | $80.81 \pm 0.61$ | $75.76 \pm 0.83$ | $88.60 \pm 0.47$ | $74.77 \pm 0.90$ | $86.88 \pm 0.59$ |
| **HFA (Ours)** | ResNet18 | $\mathbf{68.26 \pm 0.46}$ | $\mathbf{83.53 \pm 0.28}$ | $\mathbf{77.25 \pm 0.45}$ | $\mathbf{90.77 \pm 0.26}$ | $\mathbf{75.08 \pm 0.47}$ | $\mathbf{87.24 \pm 0.33}$ |

Table 2: Accuracy (%) comparisons with popular few-shot learning methods on the mini-ImageNet and tiered-ImageNet datasets. 'Optim', 'Metric', and 'Aug' denote the optimization-based, metric-based, and augmentation-based few-shot learning methods, respectively. 'Euclidean' and 'Hyperbolic' denote the algorithm is in Euclidean space and hyperbolic space, respectively. $^\star$ means that Hyper-Proto uses ResNet18 as the backbone, while the others use ResNet12.

| Method | Category | mini-ImageNet | | tiered-ImageNet | |
|---|---|---|---|---|---|
| | | 1-shot | 5-shot | 1-shot | 5-shot |
| MAML [56] | Euclidean Optim | $51.03 \pm 0.50$ | $68.26 \pm 0.47$ | $58.58 \pm 0.49$ | $71.24 \pm 0.43$ |
| L2F [57] | Euclidean Optim | $57.48 \pm 0.49$ | $74.68 \pm 0.43$ | $63.94 \pm 0.84$ | $77.61 \pm 0.41$ |
| ALFA [58] | Euclidean Optim | $60.06 \pm 0.49$ | $77.42 \pm 0.42$ | $64.43 \pm 0.49$ | $81.77 \pm 0.39$ |
| MeTAL [59] | Euclidean Optim | $59.64 \pm 0.38$ | $76.20 \pm 0.19$ | $63.89 \pm 0.43$ | $80.14 \pm 0.40$ |
| CurAML [9] | Hyperbolic Optim | $63.13 \pm 0.41$ | $81.04 \pm 0.39$ | $68.46 \pm 0.56$ | $83.84 \pm 0.40$ |
| ProtoNet [60] | Euclidean Metric | $56.52 \pm 0.45$ | $74.28 \pm 0.20$ | $53.51 \pm 0.89$ | $72.69 \pm 0.74$ |
| MetaOptNet [61] | Euclidean Metric | $62.64 \pm 0.61$ | $78.63 \pm 0.46$ | $65.99 \pm 0.72$ | $81.56 \pm 0.53$ |
| DSN [62] | Euclidean Metric | $62.64 \pm 0.66$ | $78.83 \pm 0.45$ | $66.22 \pm 0.75$ | $82.79 \pm 0.48$ |
| FEAT [8] | Euclidean Metric | $66.78 \pm 0.20$ | $82.05 \pm 0.14$ | $70.80 \pm 0.23$ | $84.79 \pm 0.16$ |
| HyperProto$^\star$ [3] | Hyperbolic Metric | $59.47 \pm 0.20$ | $76.84 \pm 0.14$ | - | - |
| w/o Aug | Hyperbolic - | $64.75 \pm 0.35$ | $79.84 \pm 0.20$ | $68.57 \pm 0.49$ | $82.22 \pm 0.39$ |
| Inf Aug | Hyperbolic Aug | $65.12 \pm 0.45$ | $80.78 \pm 0.32$ | $70.22 \pm 0.50$ | $84.51 \pm 0.37$ |
| Neural ODE + Aug 5 samples | Hyperbolic Aug | $65.54 \pm 0.46$ | $80.96 \pm 0.31$ | $69.32 \pm 0.53$ | $83.21 \pm 0.35$ |
| **HFA (Ours)** (Neural ODE + Inf Aug) | Hyperbolic Aug | $\mathbf{66.87 \pm 0.44}$ | $\mathbf{82.08 \pm 0.31}$ | $\mathbf{71.62 \pm 0.49}$ | $\mathbf{85.47 \pm 0.35}$ |

augmentation in the Euclidean feature space, while our method has better performance. For example, on the mini-ImageNet dataset using ResNet12, our method improves VFSL by 5.64% and 4.39% in the 1-shot and 5-shot tasks, respectively. The main reason is that augmentation in the hyperbolic space preserves the hierarchical structures of data and avoids the undesirable data distortion. Results on Resnet18 indicate the superiority of our method again, demonstrating the significance of data augmentation in the hyperbolic space.

We then compare HFA with popular optimization-based and metric-based few-shot learning methods on the mini-ImageNet and tiered-ImageNet datasets. Results are shown in Table 2. L2F [57], ALFA [58], MeTAL [59], and CurAML [9] are state-of-the-art optimization-based methods. Similar to them, the adaptation to new tasks of HFA is also achieved by an optimization process. Our method improves them by 3% on the 1-shot task and 1% − 2% on the 5-shot task. Compared with metric-based methods, including HyperProto [3], MetaOptNet [61], DSN [62], and FEAT [8], Our method performs competitively or even exceeds some methods. Note that CurAML and HyperProto are both few-shot learning methods in the hyperbolic space, and our better performance suggests that our data augmentation can actually improve the model in scarce data regimes.

### 5.1.2 Ablation

We conduct ablation experiments to show effectiveness of our neural ODE and the derived upper bound of the augmentation loss. We first denote 'not performing any augmentation' as 'w/o aug', where we directly use the given data to train the classifier. Then, we perform infinite augmentation, where the curvature is tuned manually, and the mean and covariance matrix are directly computed from the given data, denoted by 'Inf Aug'. Next, we use the neural ODE to estimate the distribution, and generate 5 samples for each class, denoted by 'Neural ODE + Aug 5 samples'. Finally, based on the estimated distribution, we perform infinite augmentation, denoted by 'Neural ODE + Inf Aug'. Results are shown in Table 2. We find that 'Inf Aug' has better performance than 'w/o aug',

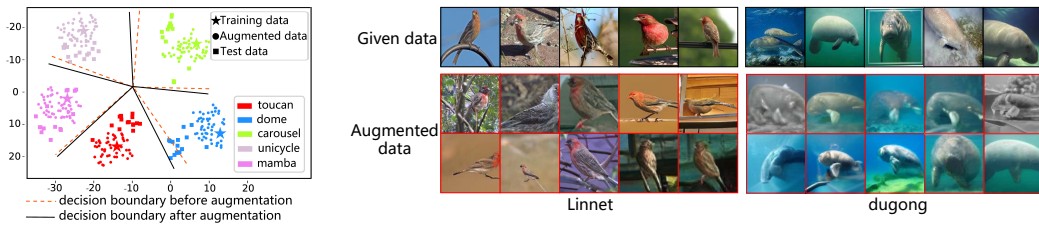

(a) Feature distribution.  (b) Corresponding images of generated features.

Figure 3: Visualization of augmentation data.

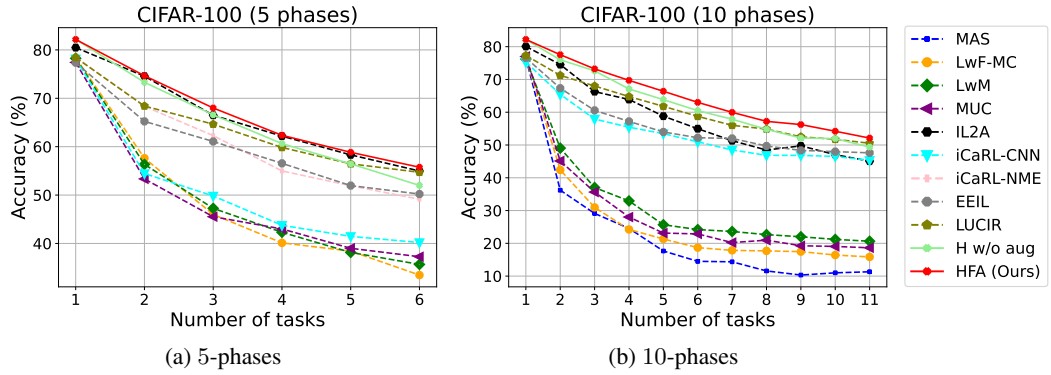

(a) 5-phases  (b) 10-phases

Figure 4: Top-1 accuracy on CIFAR-100 in continual learning.

showing that the infinite augmentation improves the generalization ability in the hyperbolic space. Using neural ODE further improves the performance on the two datasets. The reason is that directly computing the mean and covariance matrix from limited data results in imprecise distributions. In contrast, our neural ODE makes a better approximation to the real distribution.

### 5.1.3 Visualization

We first plot the estimated distribution in mini-ImageNet by sampling augmented features, where the MDS and t-SNE dimensionality reduction methods are used. Then, we use a reverse mapping algorithm [12] to search images corresponding to the augmented features for intuitive visualization. Results are shown in Figure 3. In the left panel, when training data is biased, our method well approximates the real distribution, building a robust decision boundary for classification. In the right panel, generated data has much diversities (*e.g.*, background and object posture) while preserving the class-identity.

### 5.2 Continual Learning

We use the CIFAR-100 dataset [50] with 100 classes. We evaluate HFA on two settings: $50 + 5 \times 10$ and $50 + 10 \times 5$. For example, $50 + 5 \times 10$ means that the first task contains 50 classes, and there are 10 following tasks with each one having 5 classes. We use ResNet18 as the backbone, and the model is updated by a cross-entropy loss on current data, an infinite augmentation loss on stored data, and a distillation loss. HFA stores 20 samples for each class, which is the same as replay-based methods [63, 64, 65]. Comparisons with MAS [66], LwF-MC [63], LwM [67], MUC [68], IL2A [69], iCaRL [63], EEIL [64], and LUCIR [65] are shown in Figure 4. Meanwhile, an ablation experiment where augmentation is not performed is denoted by 'H w/o aug'. Our method has better performance than replay based methods [63, 64, 65]. The reason is that our method is capable of estimating the distribution from few data, and thus leads to diverse and discriminative augmentation for stored data, making the model more reliable. In addition, our method performs better than the augmentation-based method IL2A [69]. This suggests that modeling data in the hyperbolic space is more discriminative, and performing augmentation in the hyperbolic space can further leads to a robust model.

# 6 Conclusions

In this paper, we have presented a hyperbolic feature augmentation (HFA) method that generates diverse and discriminative features to overcome overfitting in the hyperbolic space. The introduced meta-neural-ODE distribution estimation scheme can precisely approximate the real distribution in scarce data regimes. The derived upper bound of the augmentation loss makes an efficient augmentation algorithm in the hyperbolic space, which not only leads to much diversity of data but also reduces much computational load. Experiments in few-shot learning and continual learning tasks show that our method can improve the generalization of hyperbolic algorithms in scarce data regimes. In this work, we assume that data has a uniform hierarchical structure and use a single hyperbolic space for each task. Actually, real-world data may have complex hierarchical structures with varying local structures. In the future, we will study data augmentation in product manifolds to tackle such complex data.

**Acknowledgements.** This work was supported by the Natural Science Foundation of China (NSFC) under Grants No. 62172041 and No. 62176021.

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
