# Hyperbolic Feature Augmentation via Distribution Estimation and Infinite Sampling on Manifolds

**Zhi Gao**[1], **Yuwei Wu**[1,2], **Yunde Jia**[2,1], **Mehrtash Harandi**[3]

[1] Beijing Lab of Intellegent Information Technology, School of Computer Science,
Beijing Institute of Technology, China
[2] Guangdong Lab of Machine Perception and Intelligent Computing,
Shenzhen MSU-BIT University, China
[3] Department of Electrical and Computer Systems Eng.,
Monash University, and Data61, Australia
{gaozhi_2017,wuyuwei,jiayunde}@bit.edu.cn, mehrtash.harandi@monash.edu

## 1 Potential Negative Societal Impacts

Our method leads to two potential negative societal impacts.

Firstly, our method is used to generate data to train a model for human intelligence. In this case, people may heavily rely on predictions of the model, while the model may have inexplicable prediction failures, especially wrong predictions in some crucial situations.

Secondly, in the machine learning community, collecting and labeling data is a labor-intensive project. The automatic augmentation in our method may lead to a biased model.

## 2 Detailed Proof of Proposition 1

$\boldsymbol{W} = [\boldsymbol{w}_1, \cdots, \boldsymbol{w}_n]$ is the classifier, and $\boldsymbol{w}_j$ is the weight for the $j$-th classifier. $\boldsymbol{z}_{j,i}$ is the augmented feature for the $j$-th class. The loss function of infinite augmentation is

$$\mathcal{L}_\infty(\boldsymbol{W}) = \frac{1}{n} \sum_{j=1}^n \mathbb{E}_i \left[ -\log \frac{\exp\big( -d^c(\boldsymbol{z}_{j,i}, \boldsymbol{w}_j) \big)}{\sum_{j'} \exp\big( -d^c(\boldsymbol{z}_{j,i}, \boldsymbol{w}_{j'}) \big)} \right]. \tag{1}$$

Computing Eq. (1) is infeasible and difficult. Alternatively, we derive an upper bound $\overline{\mathcal{L}}_\infty(\boldsymbol{W})$ of the loss function $\mathcal{L}_\infty(\boldsymbol{W})$ for efficient augmentation, as shown in Proposition 1.

**Proposition 1.** *Given infinite augmented data $\boldsymbol{z}_{j,i} \sim \mathcal{P}_j(c, \boldsymbol{p}_j, \boldsymbol{\mu}_j, \boldsymbol{\Sigma}_j)$, an upper bound of the loss function $\mathcal{L}_\infty$ in Eq. (1) is given by*

$$\overline{\mathcal{L}}_\infty(\boldsymbol{W}) = \frac{1}{n} \sum_{j=1}^n -\log \frac{\exp\big(\boldsymbol{w}_j^\top (\widehat{\boldsymbol{p}}_j + \boldsymbol{\mu}_i)\big)}{\sum_{j'=1}^n \exp\big(\boldsymbol{w}_{j'}^\top (\widehat{\boldsymbol{p}}_j + \boldsymbol{\mu}_i) + (\boldsymbol{w}_{j'}^\top \boldsymbol{w}_{j'} - \boldsymbol{w}_j^\top \boldsymbol{w}_j) + \frac{1}{2}(\boldsymbol{w}_{j'} - \boldsymbol{w}_j)^\top \boldsymbol{\Sigma}_j (\boldsymbol{w}_{j'} - \boldsymbol{w}_j)\big)}, \tag{2}$$

*where $\boldsymbol{p}_j = \mathrm{expm}_\mathbf{0}^c(\widehat{\boldsymbol{p}}_j)$. We note that $\widehat{\boldsymbol{p}}_j$ can be directly computed in the tangent space at the origin.*

36th Conference on Neural Information Processing Systems (NeurIPS 2022).

*Proof.* The loss function of the infinite data augmentation is

$$
\begin{aligned}
\mathcal{L}_\infty(\boldsymbol{W}) &= \frac{1}{n}\sum_{j=1}^{n}\mathbb{E}_i\left[-\log\left(\frac{\exp\left(-d^c(\boldsymbol{z}_{j,i},\boldsymbol{w}_j)\right)}{\sum_{j'=1}^{n}\exp\left(-d^c(\boldsymbol{z}_{j,i},\boldsymbol{w}_{j'})\right)}\right)\right]\\
&= \frac{1}{n}\sum_{j=1}^{n}\mathbb{E}_i\left[\log\left(\frac{\sum_{j'=1}^{n}\exp\left(-d^c(\boldsymbol{z}_{j,i},\boldsymbol{w}_{j'})\right)}{\exp\left(-d^c(\boldsymbol{z}_{j,i},\boldsymbol{w}_j)\right)}\right)\right]\\
&= \frac{1}{n}\sum_{j=1}^{n}\mathbb{E}_i\left[\log\left(\sum_{j'=1}^{n}\frac{\exp\left(-d^c(\boldsymbol{z}_{j,i},\boldsymbol{w}_{j'})\right)}{\exp\left(-d^c(\boldsymbol{z}_{j,i},\boldsymbol{w}_j)\right)}\right)\right]\\
&= \frac{1}{n}\sum_{j=1}^{n}\mathbb{E}_i\left[\log\left(\sum_{j'=1}^{n}\exp\left(d^c(\boldsymbol{z}_{j,i},\boldsymbol{w}_j)-d^c(\boldsymbol{z}_{j,i},\boldsymbol{w}_{j'})\right)\right)\right]\\
&\le \frac{1}{n}\sum_{j=1}^{n}\log\left(\sum_{j'=1}^{n}\mathbb{E}_i\left[\exp\left(d^c(\boldsymbol{z}_{j,i},\boldsymbol{w}_j)-d^c(\boldsymbol{z}_{j,i},\boldsymbol{w}_{j'})\right)\right]\right),
\end{aligned}
\tag{3}
$$

where the derivation of '$\le$' in Eq. (3) follows from the Jensen's inequality $\mathbb{E}[\log(\boldsymbol{X})]\le\log\mathbb{E}[\boldsymbol{X}]$. Recall that the hyperbolic distance is

$$
d^c(\boldsymbol{x},\boldsymbol{y}) = \frac{1}{\sqrt{|c|}}\cosh^{-1}\left(1+2c\frac{\|\boldsymbol{x}-\boldsymbol{y}\|^2}{(1-c\|\boldsymbol{x}\|^2)(1-c\|\boldsymbol{y}\|^2)}\right).
\tag{4}
$$

We define a function $D = d^c(\boldsymbol{x},\boldsymbol{y})-\|\boldsymbol{x}-\boldsymbol{y}\|^2$, and denote $e = \|\boldsymbol{x}-\boldsymbol{y}\|^2$. The function $D$ can be rewritten as

$$
D = \frac{1}{\sqrt{|c|}}\cosh^{-1}\left(1+2c\frac{e}{(1-c\|\boldsymbol{x}\|^2)(1-c\|\boldsymbol{y}\|^2)}\right)-e.
\tag{5}
$$

The derivative of $D$ is given by

$$
\frac{\partial D}{\partial e} = \frac{1}{\sqrt{|c|}}\cdot\frac{q}{\sqrt{q^2e^2+2qe}}-1,
\tag{6}
$$

where $q = \frac{2c}{(1-c\|\boldsymbol{x}\|^2)(1-c\|\boldsymbol{y}\|^2)}$. Let $\frac{\partial D}{\partial e}>0$, and we obtain $e < \frac{\sqrt{1+|c|}-\sqrt{|c|}}{\sqrt{|c|}}$. Thus, given $e_1 < e_2 < \frac{\sqrt{1+|c|}-\sqrt{|c|}}{\sqrt{|c|}}$, we have $d_1^c - e_1 < d_2^c - e_2$ and $d_1^c - d_2^c < e_1 - e_2$. Here, we can reasonably assume $\|\boldsymbol{z}_{j,i}-\boldsymbol{w}_j\|^2 < \|\boldsymbol{z}_{j,i}-\boldsymbol{w}_{j'}\|^2$, since $\boldsymbol{w}_j$ is the weight of the ground-truth class of $\boldsymbol{z}_{j,i}$. Thus, if we enforce $\|\boldsymbol{z}_{j,i}-\boldsymbol{w}_{j'}\|^2 < \frac{\sqrt{1+|c|}-\sqrt{|c|}}{\sqrt{|c|}}$, we have $\|\boldsymbol{z}_{j,i},\boldsymbol{w}_j\|^2 - \|\boldsymbol{z}_{j,i},\boldsymbol{w}_{j'}\|^2 > d^c(\boldsymbol{z}_{j,i},\boldsymbol{w}_j)-d^c(\boldsymbol{z}_{j,i},\boldsymbol{w}_{j'})$, and Eq. (3) can be rewritten as

$$
\begin{aligned}
\mathcal{L}_\infty(\boldsymbol{W}) &\le \frac{1}{n}\sum_{j=1}^{n}\log\left(\sum_{j'=1}^{n}\mathbb{E}_i\left[\exp\left(d^c(\boldsymbol{z}_{j,i},\boldsymbol{w}_j)-d^c(\boldsymbol{z}_{j,i},\boldsymbol{w}_{j'})\right)\right]\right)\\
&\le \frac{1}{n}\sum_{j=1}^{n}\log\left(\sum_{j'=1}^{n}\mathbb{E}_i\left[\exp\left(\|\boldsymbol{z}_{j,i},\boldsymbol{w}_j\|^2-\|\boldsymbol{z}_{j,i},\boldsymbol{w}_{j'}\|^2\right)\right]\right)\\
&= \frac{1}{n}\sum_{j=1}^{n}\log\left(\sum_{j'=1}^{n}\mathbb{E}_i\left[\exp\left(2(\boldsymbol{w}_{j'}-\boldsymbol{w}_j)^\top\boldsymbol{z}_{j,i}+(\boldsymbol{w}_{j'}^\top\boldsymbol{w}_{j'}-\boldsymbol{w}_j^\top\boldsymbol{w}_j)\right)\right]\right).
\end{aligned}
\tag{7}
$$

We would replace $\boldsymbol{z}_{j,i}$ with the augmentation process using the parallel transport and exponential map, that is

$$\boldsymbol{z}_{j,i} = \mathrm{expm}^c_{\boldsymbol{p}_j}(\boldsymbol{v}_{j,i}) = \mathrm{expm}^c_{\boldsymbol{p}_j}(\mathrm{PT}^c_{\boldsymbol{0}\to\boldsymbol{p}_j}(\widehat{\boldsymbol{v}}_{j,i})) = \mathrm{expm}^c_{\boldsymbol{p}_j}\left(\frac{2}{\lambda^c_{\boldsymbol{p}_j}}\widehat{\boldsymbol{v}}_{j,i}\right)$$

$$= \boldsymbol{p}_j \oplus_c \left[\tanh(\sqrt{|c|}\frac{\lambda^c_{\boldsymbol{p}_j}\|\frac{2}{\lambda^c_{\boldsymbol{p}_j}}\widehat{\boldsymbol{v}}_{j,i}\|}{2})\frac{\frac{2}{\lambda^c_{\boldsymbol{p}_j}}\widehat{\boldsymbol{v}}_{j,i}}{\sqrt{|c|}\|\frac{2}{\lambda^c_{\boldsymbol{p}_j}}\widehat{\boldsymbol{v}}_{j,i}\|}\right] = \boldsymbol{p}_j \oplus_c \left[\frac{\tanh(\sqrt{|c|}\|\widehat{\boldsymbol{v}}_{j,i}\|)}{\sqrt{|c|}\|\widehat{\boldsymbol{v}}_{j,i}\|}\widehat{\boldsymbol{v}}_{j,i}\right] = \boldsymbol{p}_j \oplus_c [b\widehat{\boldsymbol{v}}_{j,i}]$$

$$= \frac{(1 - 2c\langle\boldsymbol{p}_j, b\widehat{\boldsymbol{v}}_{j,i}\rangle - c\|b\widehat{\boldsymbol{v}}_{j,i}\|^2)\boldsymbol{p}_j + (1 + c\|\boldsymbol{p}_j\|^2)b\widehat{\boldsymbol{v}}_{j,i}}{1 - 2c\langle\boldsymbol{p}_j, b\widehat{\boldsymbol{v}}_{j,i}\rangle + c^2\|\boldsymbol{p}_j\|^2\|b\widehat{\boldsymbol{v}}_{j,i}\|^2},$$

$$(8)$$

where $b = \frac{\tanh(\sqrt{|c|}\|\widehat{\boldsymbol{v}}_{j,i}\|)}{\sqrt{|c|}\|\widehat{\boldsymbol{v}}_{j,i}\|}$, and $\widehat{\boldsymbol{v}}_{j,i}$ is a Gaussian vector in the tangent space at the origin, $\widehat{\boldsymbol{v}}_{j,i} \sim \mathcal{N}(\boldsymbol{\mu}_j, \boldsymbol{\Sigma}_j)$. Here, we consider that $\boldsymbol{p}_j$ can be computed using the exponential map that projects a Euclidean vector $\widehat{\boldsymbol{p}}_j$ from the tangent space to the manifold, that is $\boldsymbol{p}_j = \mathrm{expm}^c_{\boldsymbol{0}}(\widehat{\boldsymbol{p}}_j) = \frac{\tanh(\sqrt{|c|}\|\widehat{\boldsymbol{p}}_j\|)}{\sqrt{|c|}\|\widehat{\boldsymbol{p}}_j\|}\widehat{\boldsymbol{p}}_j$. We denote $a = \frac{\tanh(\sqrt{|c|}\|\widehat{\boldsymbol{p}}_j\|)}{\sqrt{|c|}\|\widehat{\boldsymbol{p}}_j\|}$ and $\boldsymbol{p}_j = a\widehat{\boldsymbol{p}}_j$. We substitute $\boldsymbol{p}_j = a\widehat{\boldsymbol{p}}_j$ into Eq. (8) and have

$$\boldsymbol{z}_{j,i} = \frac{(a - 2ca^2b\langle\widehat{\boldsymbol{p}}_j, \widehat{\boldsymbol{v}}_{j,i}\rangle - cab^2\|\widehat{\boldsymbol{v}}_{j,i}\|^2)\widehat{\boldsymbol{p}}_j + (b + ca^2b\|\widehat{\boldsymbol{p}}_j\|^2)\widehat{\boldsymbol{v}}_{j,i}}{1 - 2cab\langle\widehat{\boldsymbol{p}}_j, \widehat{\boldsymbol{v}}_{j,i}\rangle + c^2a^2b^2\|\widehat{\boldsymbol{v}}_{j,i}\|^2\|\widehat{\boldsymbol{p}}_j\|^2}. \tag{9}$$

We substitute Eq. (9) into Eq. (7) and have

$$\mathcal{L}_\infty(\boldsymbol{W}) \le \frac{1}{n}\sum_{j=1}^n \log\left(\sum_{j'=1}^n \mathbb{E}_i\left[\exp\left(2(\boldsymbol{w}_{j'} - \boldsymbol{w}_j)^\top\boldsymbol{z}_{j,i} + (\boldsymbol{w}_{j'}^\top\boldsymbol{w}_{j'} - \boldsymbol{w}_j^\top\boldsymbol{w}_j)\right)\right]\right)$$

$$= \frac{1}{n}\sum_{j=1}^n \log\left(\sum_{j'=1}^n \mathbb{E}_i\left[\exp\left(2(\boldsymbol{w}_{j'} - \boldsymbol{w}_j)^\top\left(\frac{(a - 2ca^2b\langle\widehat{\boldsymbol{p}}_j, \widehat{\boldsymbol{v}}_{j,i}\rangle - cab^2\|\widehat{\boldsymbol{v}}_{j,i}\|^2)\widehat{\boldsymbol{p}}_j + (b + ca^2b\|\widehat{\boldsymbol{p}}_j\|^2)\widehat{\boldsymbol{v}}_{j,i}}{1 - 2cab\langle\widehat{\boldsymbol{p}}_j, \widehat{\boldsymbol{v}}_{j,i}\rangle + c^2a^2b^2\|\widehat{\boldsymbol{v}}_{j,i}\|^2\|\widehat{\boldsymbol{p}}_j\|^2}\right)\right.\right.$$

$$\left.\left. + (\boldsymbol{w}_{j'}^\top\boldsymbol{w}_{j'} - \boldsymbol{w}_j^\top\boldsymbol{w}_j)\right)\right]\right)$$

$$= \frac{1}{n}\sum_{j=1}^n \log\left(\sum_{j'=1}^n \mathbb{E}_i\left[\exp\left((\boldsymbol{w}_{j'} - \boldsymbol{w}_j)^\top\left(\frac{2(a - 2ca^2b\langle\widehat{\boldsymbol{p}}_j, \widehat{\boldsymbol{v}}_{j,i}\rangle - cab^2\|\widehat{\boldsymbol{v}}_{j,i}\|^2)}{1 - 2cab\langle\widehat{\boldsymbol{p}}_j, \widehat{\boldsymbol{v}}_{j,i}\rangle + c^2a^2b^2\|\widehat{\boldsymbol{v}}_{j,i}\|^2\|\widehat{\boldsymbol{p}}_j\|^2}\widehat{\boldsymbol{p}}_j\right.\right.\right.$$

$$\left.\left.\left. + \frac{2(b + ca^2b\|\widehat{\boldsymbol{p}}_j\|^2)}{1 - 2cab\langle\widehat{\boldsymbol{p}}_j, \widehat{\boldsymbol{v}}_{j,i}\rangle + c^2a^2b^2\|\widehat{\boldsymbol{v}}_{j,i}\|^2\|\widehat{\boldsymbol{p}}_j\|^2}\widehat{\boldsymbol{v}}_{j,i}\right) + (\boldsymbol{w}_{j'}^\top\boldsymbol{w}_{j'} - \boldsymbol{w}_j^\top\boldsymbol{w}_j)\right)\right]\right)$$

$$(10)$$

Since $\boldsymbol{w}_j$ is the weight of the ground-truth class of $\boldsymbol{z}_{j,i}$, $(\boldsymbol{w}_{j'} - \boldsymbol{w}_j)^\top\widehat{\boldsymbol{p}}_j < 0$ and $(\boldsymbol{w}_{j'} - \boldsymbol{w}_j)^\top\widehat{\boldsymbol{v}}_{j,i} < 0$ are usually satisfied. If we add some constrains to $c$, $\widehat{\boldsymbol{p}}_j$, and $\widehat{\boldsymbol{v}}_{j,i}$, such that $\frac{2(a - 2ca^2b\langle\widehat{\boldsymbol{p}}_j, \widehat{\boldsymbol{v}}_{j,i}\rangle - cab^2\|\widehat{\boldsymbol{v}}_{j,i}\|^2)}{1 - 2cab\langle\widehat{\boldsymbol{p}}_j, \widehat{\boldsymbol{v}}_{j,i}\rangle + c^2a^2b^2\|\widehat{\boldsymbol{v}}_{j,i}\|^2\|\widehat{\boldsymbol{p}}_j\|^2} > 1$ and $\frac{2(b + ca^2b\|\widehat{\boldsymbol{p}}_j\|^2)}{1 - 2cab\langle\widehat{\boldsymbol{p}}_j, \widehat{\boldsymbol{v}}_{j,i}\rangle + c^2a^2b^2\|\widehat{\boldsymbol{v}}_{j,i}\|^2\|\widehat{\boldsymbol{p}}_j\|^2} > 1$, we can remove the two terms and rewrite Eq. (10) as

$$\mathcal{L}_\infty(\boldsymbol{W}) \le \frac{1}{n}\sum_{j=1}^n \log\left(\sum_{j'=1}^n \mathbb{E}_i\left[\exp\left((\boldsymbol{w}_{j'} - \boldsymbol{w}_j)^\top\left(\frac{2(a - 2ca^2b\langle\widehat{\boldsymbol{p}}_j, \widehat{\boldsymbol{v}}_{j,i}\rangle - cab^2\|\widehat{\boldsymbol{v}}_{j,i}\|^2)}{1 - 2cab\langle\widehat{\boldsymbol{p}}_j, \widehat{\boldsymbol{v}}_{j,i}\rangle + c^2a^2b^2\|\widehat{\boldsymbol{v}}_{j,i}\|^2\|\widehat{\boldsymbol{p}}_j\|^2}\widehat{\boldsymbol{p}}_j\right.\right.\right.$$

$$\left.\left.\left. + \frac{2(b + ca^2b\|\widehat{\boldsymbol{p}}_j\|^2)}{1 - 2cab\langle\widehat{\boldsymbol{p}}_j, \widehat{\boldsymbol{v}}_{j,i}\rangle + c^2a^2b^2\|\widehat{\boldsymbol{v}}_{j,i}\|^2\|\widehat{\boldsymbol{p}}_j\|^2}\widehat{\boldsymbol{v}}_{j,i}\right) + (\boldsymbol{w}_{j'}^\top\boldsymbol{w}_{j'} - \boldsymbol{w}_j^\top\boldsymbol{w}_j)\right)\right]\right)$$

$$\le \frac{1}{n}\sum_{j=1}^n \log\left(\sum_{j'=1}^n \mathbb{E}_i\left[\exp\left((\boldsymbol{w}_{j'} - \boldsymbol{w}_j)^\top(\widehat{\boldsymbol{p}}_j + \widehat{\boldsymbol{v}}_{j,i}) + (\boldsymbol{w}_{j'}^\top\boldsymbol{w}_{j'} - \boldsymbol{w}_j^\top\boldsymbol{w}_j)\right)\right]\right).$$

$$(11)$$

Considering $\widehat{\boldsymbol{v}}_{j,i} \sim \mathcal{N}(\boldsymbol{\mu}_j, \boldsymbol{\Sigma}_j)$, the term $(\boldsymbol{w}_{j'} - \boldsymbol{w}_j)^\top (\widehat{\boldsymbol{p}}_j + \widehat{\boldsymbol{v}}_{j,i}) + (\boldsymbol{w}_{j'}^\top \boldsymbol{w}_{j'} - \boldsymbol{w}_j^\top \boldsymbol{w}_j)$ in Eq. (11) is a Gaussian vector as well,

$$
\begin{aligned}
&(\boldsymbol{w}_{j'} - \boldsymbol{w}_j)^\top (\widehat{\boldsymbol{p}}_j + \widehat{\boldsymbol{v}}_{j,i}) + (\boldsymbol{w}_{j'}^\top \boldsymbol{w}_{j'} - \boldsymbol{w}_j^\top \boldsymbol{w}_j) \\
&\sim \mathcal{N}\Big( (\boldsymbol{w}_{j'} - \boldsymbol{w}_j)^\top (\widehat{\boldsymbol{p}}_j + \boldsymbol{\mu}_j) + (\boldsymbol{w}_{j'}^\top \boldsymbol{w}_{j'} - \boldsymbol{w}_j^\top \boldsymbol{w}_j), (\boldsymbol{w}_{j'} - \boldsymbol{w}_j)^\top \boldsymbol{\Sigma}_j (\boldsymbol{w}_{j'} - \boldsymbol{w}_j) \Big).
\end{aligned}
\tag{12}
$$

Based on the moment-generating function, $\mathbb{E}[\exp(\boldsymbol{x})] = \exp(\boldsymbol{\mu} + \frac{1}{2}\boldsymbol{\Sigma})$, $\boldsymbol{x} \sim \mathcal{N}(\boldsymbol{\mu}, \boldsymbol{\Sigma})$, we can rewrite Eq. (11) as

$$
\begin{aligned}
\mathcal{L}_\infty(\boldsymbol{W}) &\le \frac{1}{n} \sum_{j=1}^n \log \left( \sum_{j'=1}^n \mathbb{E}_i \left[ \exp\left( (\boldsymbol{w}_{j'} - \boldsymbol{w}_j)^\top (\widehat{\boldsymbol{p}}_j + \widehat{\boldsymbol{v}}_{j,i}) + (\boldsymbol{w}_{j'}^\top \boldsymbol{w}_{j'} - \boldsymbol{w}_j^\top \boldsymbol{w}_j) \right) \right] \right) \\
&= \frac{1}{n} \sum_{j=1}^n \log \left( \sum_{j'=1}^n \exp \left( (\boldsymbol{w}_{j'} - \boldsymbol{w}_j)^\top (\widehat{\boldsymbol{p}}_j + \boldsymbol{\mu}_j) + (\boldsymbol{w}_{j'}^\top \boldsymbol{w}_{j'} - \boldsymbol{w}_j^\top \boldsymbol{w}_j) + \frac{1}{2}(\boldsymbol{w}_{j'} - \boldsymbol{w}_j)^\top \boldsymbol{\Sigma}_j (\boldsymbol{w}_{j'} - \boldsymbol{w}_j) \right) \right) \\
&= \frac{1}{n} \sum_{j=1}^n -\log \frac{\exp\left( \boldsymbol{w}_j (\widehat{\boldsymbol{p}}_j + \boldsymbol{\mu}_j) \right)}{\exp\left( \boldsymbol{w}_{j'}(\widehat{\boldsymbol{p}}_j + \boldsymbol{\mu}_j) + (\boldsymbol{w}_{j'}^\top \boldsymbol{w}_{j'} - \boldsymbol{w}_j^\top \boldsymbol{w}_j) + \frac{1}{2}(\boldsymbol{w}_{j'} - \boldsymbol{w}_j)^\top \boldsymbol{\Sigma}_j (\boldsymbol{w}_{j'} - \boldsymbol{w}_j) \right)} \\
&= \overline{\mathcal{L}}_\infty(\boldsymbol{W}).
\end{aligned}
\tag{13}
$$

**Assumption.** In this derivation, we have two assumptions, that is, (1) $\|\boldsymbol{z}_{j,i} - \boldsymbol{w}_j\|^2 < \|\boldsymbol{z}_{j,i} - \boldsymbol{w}_{j'}\|^2 < \frac{\sqrt{1+|c|} - \sqrt{|c|}}{\sqrt{|c|}}$ in Eq. (7), and (2) $\frac{2(a - 2ca^2 b\langle \widehat{\boldsymbol{p}}_j, \widehat{\boldsymbol{v}}_{j,i}\rangle - cab^2\|\widehat{\boldsymbol{v}}_{j,i}\|^2)}{1 - 2cab\langle \widehat{\boldsymbol{p}}_j, \widehat{\boldsymbol{v}}_{j,i}\rangle + c^2 a^2 b^2 \|\widehat{\boldsymbol{v}}_{j,i}\|^2 \|\widehat{\boldsymbol{p}}_j\|^2} > 1$ and $\frac{2(b + ca^2 b\|\widehat{\boldsymbol{p}}_j\|^2)}{1 - 2cab\langle \widehat{\boldsymbol{p}}_j, \widehat{\boldsymbol{v}}_{j,i}\rangle + c^2 a^2 b^2 \|\widehat{\boldsymbol{v}}_{j,i}\|^2 \|\widehat{\boldsymbol{p}}_j\|^2} > 1$ in Eq. (11).

For the assumption (1), we initialize the weight of the classifier as $\boldsymbol{w}_j = \boldsymbol{p}_j$ to help $\|\boldsymbol{z}_{j,i} - \boldsymbol{w}_j\|^2 < \|\boldsymbol{z}_{j,i} - \boldsymbol{w}_{j'}\|^2$. $\boldsymbol{z}_{j,i}$ is obtained by $\boldsymbol{z}_{j,i} = \frac{(a - 2ca^2 b\langle \widehat{\boldsymbol{p}}_j, \widehat{\boldsymbol{v}}_{j,i}\rangle - cab^2\|\widehat{\boldsymbol{v}}_{j,i}\|^2)\widehat{\boldsymbol{p}}_j + (b + ca^2 b\|\widehat{\boldsymbol{p}}_j\|^2)\widehat{\boldsymbol{v}}_{j,i}}{1 - 2cab\langle \widehat{\boldsymbol{p}}_j, \widehat{\boldsymbol{v}}_{j,i}\rangle + c^2 a^2 b^2 \|\widehat{\boldsymbol{v}}_{j,i}\|^2 \|\widehat{\boldsymbol{p}}_j\|^2}$, and $\boldsymbol{w}_j$ is obtained by $\boldsymbol{w}_j = \frac{\tanh(\sqrt{|c|}\|\widehat{\boldsymbol{p}}_j\|)}{\sqrt{|c|}\|\widehat{\boldsymbol{p}}_j\|}\widehat{\boldsymbol{p}}_j = a\widehat{\boldsymbol{p}}_j$. $\|\boldsymbol{z}_{j,i} - \boldsymbol{w}_{j'}\|^2 < \frac{\sqrt{1+|c|} - \sqrt{|c|}}{\sqrt{|c|}}$ can be achieved by enforcing

$$
\begin{cases}
\|\boldsymbol{z}_{j,i}\|^2 < \left( \dfrac{(a - 2ca^2 b\langle \widehat{\boldsymbol{p}}_j, \widehat{\boldsymbol{v}}_{j,i}\rangle - cab^2\|\widehat{\boldsymbol{v}}_{j,i}\|^2)\|\widehat{\boldsymbol{p}}_j\| + (b + ca^2 b\|\widehat{\boldsymbol{p}}_j\|^2)\|\widehat{\boldsymbol{v}}_{j,i}\|}{1 - 2cab\langle \widehat{\boldsymbol{p}}_j, \widehat{\boldsymbol{v}}_{j,i}\rangle + c^2 a^2 b^2 \|\widehat{\boldsymbol{v}}_{j,i}\|^2 \|\widehat{\boldsymbol{p}}_j\|^2} \right)^2 < \dfrac{\sqrt{1+|c|} - \sqrt{|c|}}{4\sqrt{|c|}} \\[3mm]
\|\boldsymbol{w}_j\|^2 = a^2\|\widehat{\boldsymbol{p}}_j\|^2 < \dfrac{\sqrt{1+|c|} - \sqrt{|c|}}{4\sqrt{|c|}}
\end{cases}
.
$$

$$
\tag{14}
$$

When we enforce $\|\widehat{\boldsymbol{p}}_j\| < 0.25$ and $\|\boldsymbol{\mu}_j\| < 0.25$, and curvature is generated in the range of $[-0.5, 0]$, Eq. (14) can be satisfied. Meanwhile, the constraints of $\widehat{\boldsymbol{p}}_j$, $\boldsymbol{\mu}_j$, and $c$ make $\sqrt{|c|}\|\widehat{\boldsymbol{p}}_j\| < 0.2$, $\sqrt{|c|}\|\widehat{\boldsymbol{v}}_{j,i}\| < 0.2$, $a > 0.98$, and $b > 0.98$, through which the assumption (2) is also satisfied, *i.e.*, $\frac{2(a - 2ca^2 b\langle \widehat{\boldsymbol{p}}_j, \widehat{\boldsymbol{v}}_{j,i}\rangle - cab^2\|\widehat{\boldsymbol{v}}_{j,i}\|^2)}{1 - 2cab\langle \widehat{\boldsymbol{p}}_j, \widehat{\boldsymbol{v}}_{j,i}\rangle + c^2 a^2 b^2 \|\widehat{\boldsymbol{v}}_{j,i}\|^2 \|\widehat{\boldsymbol{p}}_j\|^2} > 1$, and $\frac{2(b + ca^2 b\|\widehat{\boldsymbol{p}}_j\|^2)}{1 - 2cab\langle \widehat{\boldsymbol{p}}_j, \widehat{\boldsymbol{v}}_{j,i}\rangle + c^2 a^2 b^2 \|\widehat{\boldsymbol{v}}_{j,i}\|^2 \|\widehat{\boldsymbol{p}}_j\|^2} > 1$. $\qquad\square$

## 3 Experimental Details

### 3.1 Few-shot Learning

**Dataset.** We conduct experiments on four popular datasets, namely mini-ImageNet [1], tiered-ImageNet [2], CUB [3], and CIFAR-FS dataset [4]. The mini-ImageNet dataset contains 100 classes from the ImageNet dataset, and each class has 600 images. We split the 100 classes into 64, 16, and 20 classes for training, validation, and testing, respectively. The tiered-ImageNet dataset has 779165 images from 608 classes totally, where 351, 97, and 160 classes are used for training, validation, and testing, respectively. The CUB dataset is a fine-grained image dataset that contains 200 bird classes and 11788 images totally. Following the protocol of [5], we utilize 100, 50, and 50 classes for training, validation, and testing, respectively. We crop bird regions from images with the given bounding boxes before training. CIFAR-FS is a few-shot learning dataset derived from CIFAR-100.

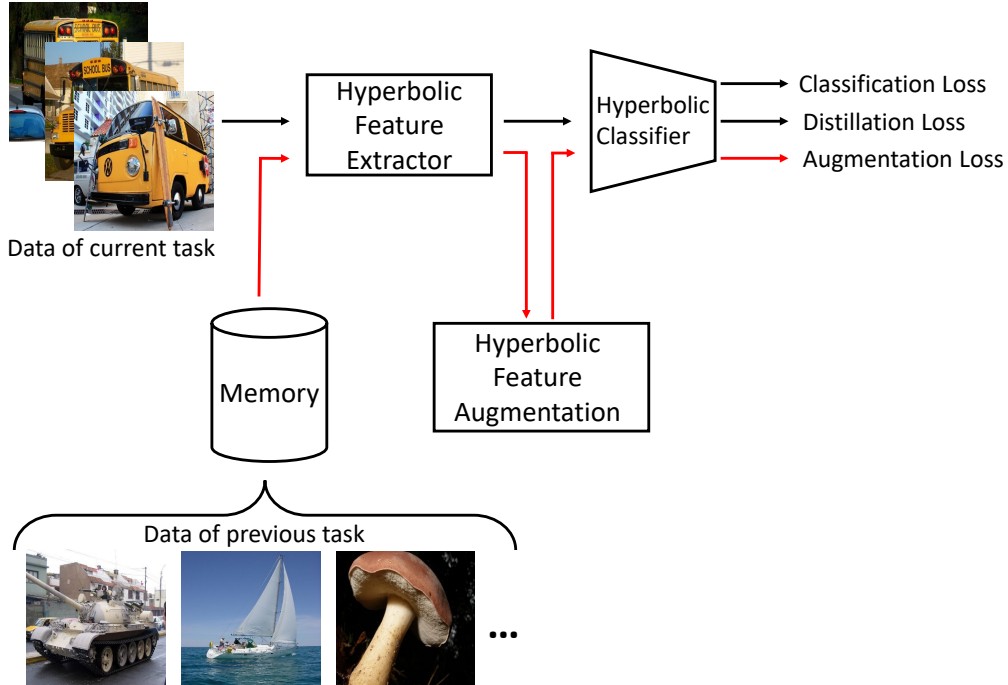

Figure 1: The overview of the proposed HFA method in the continual learning task.

It contains 100 classes, where 64, 16, and 20 classes are used for training, validation, and testing, respectively. Each class has 600 images. When the ResNet12 backbone is used, the image size for the mini-ImageNet, tiered-ImageNet, and CUB datasets is $84 \times 84$, and the image size for the CIFAR-FS dataset is $32 \times 32$. When the ResNet18 backbone is used, the image size for the mini-ImageNet, tiered-ImageNet, CUB, and CIFAR-FS datasets is $224 \times 224$.

**Model details.** We regard features from the backbones are located in the tangent space of the hyperbolic space at the origin. We add an exponential map on the top of the backbone to project features from the tangent space to the hyperbolic space. The two backbones are pre-trained on the training set, using the cross-entropy loss. Then, the two backbones are fixed in our meta-learning process.

**Hyperaparameter.** In the pre-training stage, we train the two backbones over 120 epochs with the SGD optimizer. We set the learning rate was as 0.01 in the pre-training stage, decay the learning rate per 40 epochs, and the decay rate is set as 0.1. We remove the last fully-connected layer and the softmax layer of the pre-trained model, and the rest layers are used in our feature extractor. In the meta-learning stage, we train the three gradient flow networks over 4000 episodes. We use the Adam optimizer in the meta-training stage, where the learning rate is 0.001. We decay the learning rate per 2000 episodes in the meta-training stage, and the decay rate is 0.1. We set the weight decay as 0.0005 for the feature extractor.

In the few-shot learning experiment, the ratio of $\mathcal{D}_t$ to $\mathcal{D}_v$ follows the standard protocol. In the 1-shot setting, the ratio is $1 : 15$, that is, we sample one sample as the training data and 15 samples as the validation data in each task. In the 5-shot setting, the ratio is $1 : 3$, that is, we sample 5 sample as the training data and 15 samples as the validation data in each task.

The initial value of the curvature $c$ is set as $0$.

## 3.2 Continual Learning

**Dataset.** We use the CIFAR-100 dataset with 100 classes for the continual learning task, where the image size is $32 \times 32$. We evaluate HFA on two settings: $50 + 5 \times 10$ and $50 + 10 \times 5$. For example, $50 + 5 \times 10$ means that the first task contains 50 classes, and there are 10 following tasks with each one having 5 classes.

**Model details.** The overview of HFA in the continual learning task is shown in Figure 1. Our model contains four modules: a hyperbolic feature extractor $f(\cdot)$, a hyperbolic classifier $C(\cdot)$ using the hyperbolic distance, a memory module, and a hyperbolic feature augmentation algorithm with three gradient flow networks $F_1$, $F_2$, and $F_3$. In the continual learning task, we apply ResNet18 as the feature extractor. Similar to the few-shot learning task, we regard features extracted from ResNet18 are located in the tangent space at the origin. We then use an exponential map to project features to the hyperbolic space. The hyperbolic classifier is a distance-based classifier. For the memory module, we save 20 samples for each class, following existing reply-based continual learning methods [6, 7, 8].

**Training details.** In continual learning, sequence of the $t+1$ task are given with data $D^0, D^1, \cdots, D^t$. The training process of HFA in continual learning has three stages.

(1) In the first stage, the first task is given with data $D^0$. We use $D^0$ to train the hyperbolic feature extractor $f(\cdot)$ and the hyperbolic classifier $C(\cdot)$, using the cross-entropy loss,

$$\mathcal{L}_c^0 = \mathbb{E}_{\boldsymbol{I}^0 \in D^0}\left[-\sum_{j=1}^n \mathbb{1}_{y=j} \log \frac{\exp\left(C(f(\boldsymbol{I}^0))\right)}{\sum_{j'=1}^n \exp\left(C(f(\boldsymbol{I}^0))\right)}\right]. \tag{15}$$

where $\boldsymbol{I}^0 \in D^0$ is the image belonging to the first task, and $y$ is its label.

(2) Based on the trained hyperbolic feature extractor, we train the three gradient flow networks via meta-learning with a bi-level optimization process. In each inner-loop, $D^0$ is regard as the base data, and we randomly sample several data from $D^0$ as the training data $D_t^0$, and randomly select some data from $D^0$ as the validation data $D_v^0$. We initialize the classifier as the mean of $D_t^0$, and update it by the derived upper bound $\overline{\mathcal{L}}_\infty$ of the augmentation loss. In the outer-loop, we update the three gradient flow networks by minimizing the classification loss of the update classifier on the validation data $D_v^0$.

(3) In the third stage, we train the model in the following $1 - t$ tasks. We train the hyperbolic feature extractor and the hyperbolic classifier by minimizing the classification loss $\mathcal{L}_c$ on data of the current task, a distillation loss $\mathcal{L}_d$ on data of the current task, and the derived upper bound $\overline{\mathcal{L}}_\infty$ of the augmentation loss on data of previous tasks stored in the memory module. The classification loss is given by

$$\mathcal{L}_c = \mathbb{E}_{\boldsymbol{I}^t \in D^t}\left[-\sum_{j=1}^n \mathbb{1}_{y=j} \log \frac{\exp\left(C(f(\boldsymbol{I}^t))\right)}{\sum_{j'=1}^n \exp\left(C(f(\boldsymbol{I}^t))\right)}\right]. \tag{16}$$

Note that, the distillation loss is defined in the hyperbolic space. We denote the feature extractor of the previous task as $f'(\cdot)$, the current feature extractor as $f(\cdot)$, and the memory module as $\mathcal{B}$. The distillation loss is

$$\mathcal{L}_d = \mathbb{E}_{\boldsymbol{I} \in \mathcal{B}}[d^c(f'(\boldsymbol{I}), f(\boldsymbol{I}))]. \tag{17}$$

In summary, the loss function in the third stage is given by,

$$\mathcal{J} = \mathcal{L}_c + \alpha_1 \overline{\mathcal{L}}_\infty + \alpha_2 \mathcal{L}_d, \tag{18}$$

where $\alpha_1$ and $\alpha_2$ are the trade-off parameters. The whole training process of continual learning is summarized in Algorithm 1.

**Hyperparameter.** We use the SGD optimizer in the three training stage. We train the model over 100 epochs in the first stage, 100 epoch in the second stage, and 5 epochs in the third stage. The learning rate is 0.001 in the first stage, 0.01 in the second stage, and 0.0003 in the third stage. The weight decay for the backbone is 0.0005. We save 20 samples for each class in the memory module. In the second stage, we sample 32 samples as the training data, and 32 samples as the validation data. In the third stage, we sample 5 samples for each class in the memory module to estimate the distribution. We set $\alpha_1 = 10$ and $\alpha_2 = 10$ in the third stage.

In the continual learning task, the ratio of $\mathcal{D}_t$ to $\mathcal{D}_v$ is $1 : 1$.

The initial value of the curvature $c$ is set as $0$.

**Algorithm 1** Our training process in continual learning.

---

**Input:** Sequence of tasks with data $D^0, D^1, \cdots, D^t$.
**Output:** A model for continual learning, which has a hyperbolic feature extractor $f(\cdot)$, a hyperbolic
   classifier $C(\cdot)$, and three gradient flow networks $F_1$, $F_2$, and $F_3$.
 1: **while** not converged **do**
 2:    We train the $f(\cdot)$ and $C(\cdot)$ by minimizing the classification loss in Eq. (15) on $D^0$.
 3: **end while**
 4: **while** not converged **do**
 5:    Randomly select few training data $\mathcal{D}_t^0$ from $\mathcal{D}^0$, and randomly select some data from $\mathcal{D}^0$ as
       the validation data $\mathcal{D}_v^0$.
 6:    Estimate the distribution for each class in $\mathcal{D}_t^0$ by using $F_1$, $F_2$, and $F_3$.
 7:    **while** not converged **do**
 8:       Train the hyperbolic classifier $C(\cdot)$ using the estimated distribution by minimizing $\overline{\mathcal{L}}$.
 9:    **end while**
10:    Update $F_1$, $F_2$, and $F_3$ by minimizing the classification loss of the updated classifier using the
       validation data $\mathcal{D}_v^0$.
11: **end while**
12: Randomly select data from $\mathcal{D}^0$ and store them in $\mathcal{B}$.
13: **for** $s = 1$ to $t$ **do**
14:    **while** not converged **do**
15:       We train the $f(\cdot)$ and $C(\cdot)$ by minimizing the loss function in Eq. (18) on $D^s$.
16:    **end while**
17:    Randomly select data from $\mathcal{D}^s$ and store them in $\mathcal{B}$
18: **end for**

---

## 3.3 Reverse mapping algorithm

Inspired by the work [9], we utilize a reverse mapping algorithm to convert augmented features to
images. The overview of the reverse mapping algorithm is shown in Figure 2.

This algorithm has two networks: a pre-trained image generator $\mathcal{G}$ to generate fake images, and a
pre-trained feature extractor network $f$ to extract hyperbolic features of images. In our experiments,
we visualize images for the mini-ImageNet dataset, and thus we utilize the image generator from
the work [10] as $\mathcal{G}$, which is pre-trained on ImageNet, and we pre-train $f$ on the mini-ImageNet
dataset. We use $z \in \mathbb{R}^d$ to denote a random noize vector, and use $I$ to denote a real image from the
mini-ImageNet dataset. The reverse mapping algorithm has three steps.

(1) We feed $z$ into $\mathcal{G}$ to generate a fake image $\mathcal{G}(z)$. Then, we feed $\mathcal{G}(z)$ and the real image $I$ into $f$
to extract their hyperbolic features $f(\mathcal{G}(z))$ and $f(I)$, respectively. Here, we update the noize $z$ to
find its optimal value that corresponds to the real image $I$ by minimizing the following loss function,

$$z^* = \arg \min_{z} d^c\Big( f(I) - f(\mathcal{G}(z)) \Big) + \alpha \|I - \mathcal{G}(z)\|^2, \tag{19}$$

where $\alpha$ is the trade-off hyparameter, and we obtain the optimal value $z^*$.

(2) We perform our augmentation algorithm on $f(I)$ and obtain the augmented feature $\widehat{f}(I)$. We
initialize a new vector $z = z^*$, and update $z$ to find the optimal value that corresponds to the
augmented feature $\widehat{f}(I)$ by minimizing the following loss function,

$$z^{*'} = \arg \min_{z} d^c\Big( f(\mathcal{G}(z)), \widehat{f}(I) \Big). \tag{20}$$

After the optimization, we obtain $z^{*'}$.

(3) We generate images that corresponds to the augmented feature $\widehat{f}(I)$ by

$$\widehat{I} = \mathcal{G}(z^{*'}). \tag{21}$$

## 3.4 Experimental configuration

We use an Inter(R) Core(TM) i9-10900X 3.7GHz CPU, a GeForce RTX 3090 GPU, and 128GB
RAM to conduct experiments. We use CUDA 11.3, Python 3.6, and Pytorch 1.10.

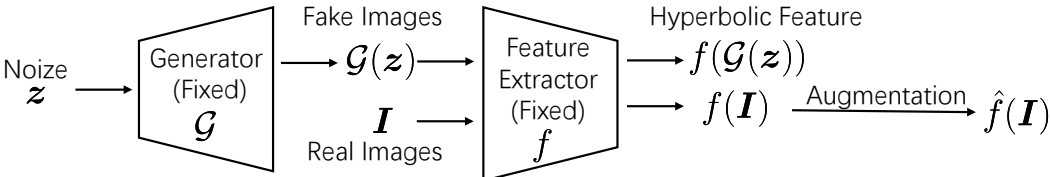

Figure 2: The overview of the reverse mapping algorithm.

### 3.5 Analysis of the Bi-level Optimization

In the bi-level optimization process, hyper-parameters include the learning rate in the two loops, the choice of the optimizer in the two loops, and the number of iterative steps in the two loops. The inner-loop optimization plays an important role in efficiently solving the bi-level optimization. An advanced optimizer and a large number of iteration steps in the inner-loop may lead to a good inner-loop solution, benefiting the outer-loop solution. However, this may increase resource consumption and cause instabilities in the gradient (exploding gradient issue). Using a simpler optimizer and a small number of iteration steps can reduce the resource consumption, but it may lead to a biased solution of the inner-loop, resulting in poor performance.

In the implementation, we recommend using a simple gradient descent optimizer in the inner loop to reduce computational complexity, and using the Adam optimizer in the outer-loop to reduce the bias of optimization. As for the number of iterative steps, we empirically observe that changing the number of iterations of the inner loop in the range $[5, 20]$ does not deteriorate the performance, and setting the number of iterations in the outer-loop larger than 1000 make the model converge. In terms of learning rates of the two loops, we change them in the range $[0.001, 0.0001]$, and all get good performance. This shows the robustness of our method to hyper-parameters if they are chosen within a suitable interval.

## 4 Extra Experimental Results

### 4.1 Loss Curves

We draw a loss curve in the few-shot learning task to show the overfitting problem when limited data is given. We use the mini-ImageNet dataset, and experiments are conducted on the 5-shot 5-way task. We first draw the loss curves of 'not augmentation', where the hyperbolic model is trained by using the original given data, and the training and test losses are denoted by 'Original Train' and 'Original Test'. Then, we draw the loss curves of using infinite augmentation via the derived upper bound, where the training and test losses are denoted by 'Ours Train' and 'Ours Test'. Here, we draw the curves in 5 optimization steps. Results are shown in Figure 3. The loss curves are the mean of losses over 1000 tasks. We have the following two conclusions. (1) Using the limited data to train a hyperbolic model actually results in the overfitting problem. After the 5 optimization steps, the training loss is about $1.43$, while the test loss is about $1.56$. (2) Our method solves the overfitting problem by data augmentation. After 5 optimization steps, our training loss is about $1.37$, and our test loss is about $1.39$.

### 4.2 Graph Node Classification

We add an experiment on the graph node classification task using the DISEASE [11], CORA [12], and PUBMED [13] datasets. Concretely, we use HGCN [14] as the baseline. We add HFA between the graph convolutional network and the node classifier, and perform augmentation for node features. We train the gradient flow networks to estimate data distributions of node features, and use augmented features to train the node classifier. We report the average F1-score ($\%$) with the standard deviation on 10 random experiments. Results are shown in Table 1. This experiment shows that our method leads to improvements for graph data as well. For example, on the DISEASE dataset, the F1 score of HGCN is $74.5\%$. In contrast, HGCN+HFA achieves $78.0\%$, $3.5\%$ higher than HGCN.

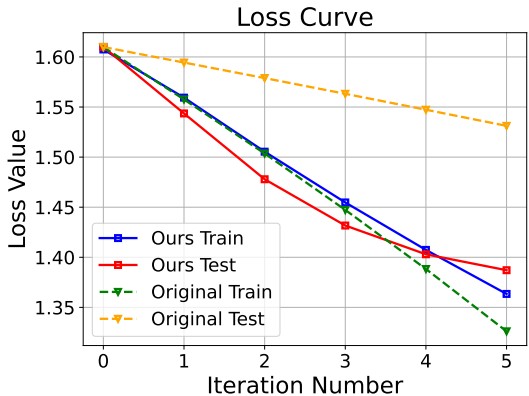

Figure 3: Loss curves of using original data and our infinite augmented data in the mini-ImageNet dataset. For the original data, 5 images are given in each class.

Table 1: Node Classification Results.

| Method | DISEASE | CORA | PUBMED |
|---|---|---|---|
| HGCN [14] | $74.5 \pm 0.9$ | $79.9 \pm 0.6$ | $80.3 \pm 0.3$ |
| HGCN + HFA | $\mathbf{78.0 \pm 0.7}$ | $\mathbf{81.1 \pm 0.7}$ | $\mathbf{81.5 \pm 0.2}$ |

### 4.3 Evaluation of Estimated Distribution

We conduct a toy experiment to evaluate the capability of neural ODE, where we train a neural ODE module to estimate Gaussian distributions in the Euclidean space. We randomly generate some vectors and matrices as the means and covariance matrices of Gaussian distributions, and sample few data from these distributions to simulate the setting of low data regime. Then, the neural ODE is trained to estimate the means and covariance matrices based on the sampled data. Concretely, we generate 5 samples from each distribution and use the MSE loss to train the neural ODE. We compare the neural ODE with directly computing mean and covariance matrices from given data, where the framework is presented in Figure 4. Results are evaluated over 1000 distributions, as shown in Table 2. Using the neural-ODE module leads to a more precise distribution estimation in the low data regime. For estimating the mean, the error of using neural ODE is $2.96 \times 10^4$ that is one percent of the error of directly computing. For estimating the covariance matrix, the error of using neural ODE is $5.23 \times 10^2$, far less than the error of direct computing, which is $2.83 \times 10^{11}$.

We further conduct an experiment to demonstrate this point. We randomly generate some hyperbolic wrapped normal distributions, and sample few data from them (samples from each distribution), where the label of a sample is set by its corresponding distribution. Then we train the gradient flow networks via the bi-level optimization. Finally, we use the gradient flow networks to recover unseen hyperbolic wrapped normal distributions. Results are shown in Table 3. Results are that the MSE

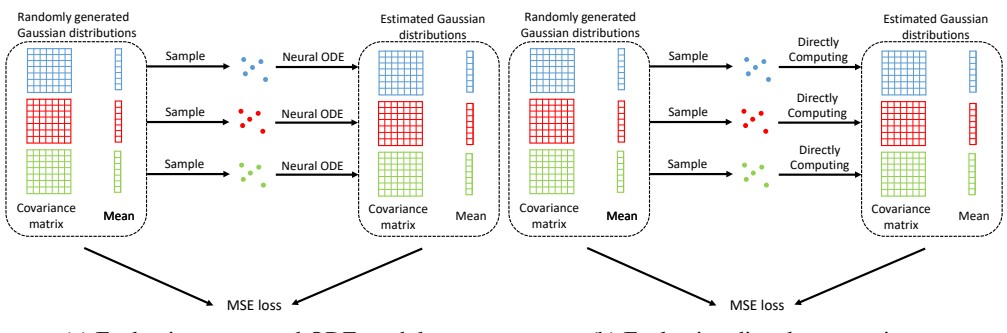

(a) Evaluating our neural ODE module.          (b) Evaluating directly computing.

Figure 4: Framework of a toy experiment for evaluation of estimated distributions.

Table 2: MSE between the estimated distribution and ground truth over 1000 Gaussian distributions.

| Method | Loss of Mean | Loss of Covariance |
|---|---|---|
| Directly computing | $2.41 \times 10^6$ | $2.83 \times 10^{11}$ |
| Neural ODE | $2.96 \times 10^4$ | $5.23 \times 10^2$ |

Table 3: MSE between the estimated distribution and ground truth over 1000 hyperbolic wrapped normal distributions.

| Method | Loss of Mean | Loss of Covariance | Loss of Curvature |
|---|---|---|---|
| Directly computing | $9.08 \times 10^3$ | $5.12\times^5$ | $9.05$ |
| Neural ODE | $41.69$ | $5.17 \times 10^3$ | $0.42$ |

errors of directly computing distribution parameters from given data are $9.08 \times 10^3$, $5.12\times^5$, and $9.05$ for the mean, covariance matrix, and curvature, respectively. In contrast, the MSE errors of using the trained gradient flow networks are $41.69$, $5.17 \times 10^3$, and $0.42$ for the mean, covariance matrix, and curvature, respectively. These results show that the trained gradient flow networks are capable of generalizing to unseen data in the scarce data setting.

## 4.4 Efficiency

In this section, we conduct an ablation experiment in the few-shot learning task to show the efficiency of the derived upper bound of the original augmentation loss. We evaluate the wall-clock time of training the hyperbolic classifier using the derived upper bound with implicitly augmented features and the original augmentation loss with explicitly augmented features. Results are shown in Table 4, where $\overline{\mathcal{L}}_\infty$ means using the upper bound. $\mathcal{L}, m = 5$ means using the original augmentation loss to train the classifier, and 5 features are generated for each class. The time is evaluated over 1000 1-shot 5-way and 1000 5-shot 5-way tasks. The derived upper bound significantly reduces the time consumption of augmentation and leads to better performance, making an efficient hyperbolic augmentation algorithm. This is because it does not need to explicitly generate features, and avoid complex hyperbolic operations.

Table 4: Time (seconds) and accuracy (%) of feature augmentation on the mini-ImageNet dataset.

| Method | 1-shot time | 1-shot time | 1-shot accuracy | 1-shot time |
|---|---|---|---|---|
| $\mathcal{L}, m = 5$ | 85 | 86 | $65.54 \pm 0.46$ | $80.96 \pm 0.31$ |
| $\mathcal{L}, m = 10$ | 98 | 101 | $65.60 \pm 0.44$ | $81.11 \pm 0.20$ |
| $\mathcal{L}, m = 20$ | 103 | 110 | $65.75 \pm 0.44$ | $81.20 \pm 0.31$ |
| $\mathcal{L}, m = 50$ | 111 | 114 | $65.95 \pm 0.46$ | $81.29 \pm 0.30$ |
| $\overline{\mathcal{L}}_\infty$ | **28** | **36** | $\mathbf{66.87 \pm 0.44}$ | $\mathbf{82.08 \pm 0.31}$ |

## 4.5 More Visualization Results

### 4.5.1 Distribution Visualization

We draw the estimated distribution in the mini-ImageNet dataset by sampling augmented features, where the MDS and t-SNE dimensionality reduction methods are used. Results are shown in Figure 5, 6, 7, 8, 9, 10, 11, 12, 13, 14. From the figures, we have two conclusions. (1) By generating more features, our model has larger inter-class distances and smaller intra-class distances. (2) When training data is biased, our method well approximates the real distribution, helping building a robust decision boundary.

### 4.5.2 Corresponding Images of Generated Features

We search images corresponding to the augmented features by the reverse mapping algorithm. Results are shown in Figure 15, 16, 17, 18, 19, 20, 21, 22, 23, 24. We can conclude that the generated features have much diversities. For example, in Figure 23, generated images of dogs have different postures and backgrounds.

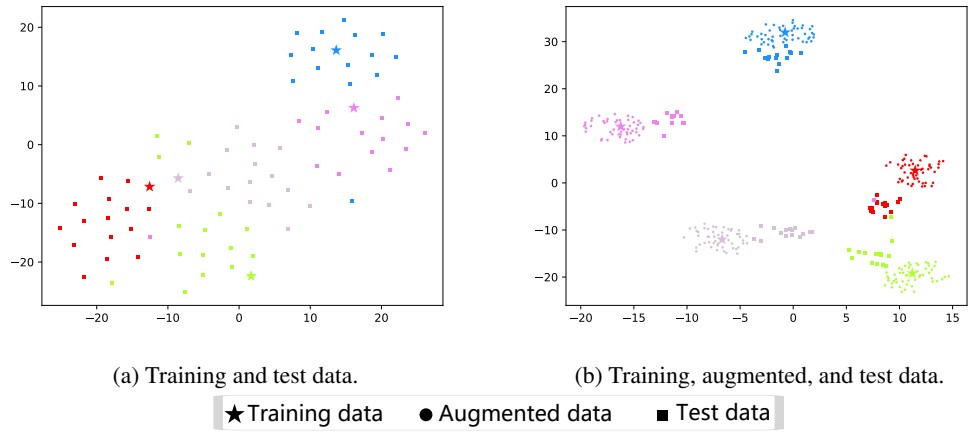

(a) Training and test data.      (b) Training, augmented, and test data.

★ Training data    ● Augmented data    ■ Test data

Figure 5: An example of feature distributions of one few-shot task.

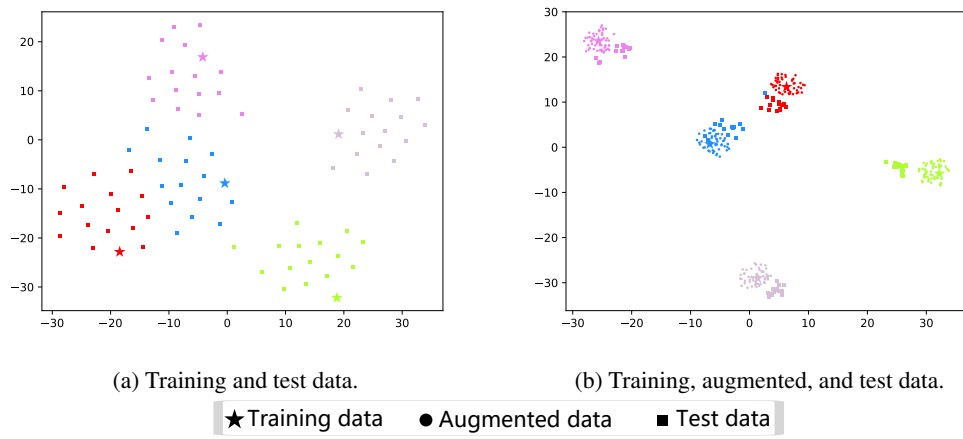

(a) Training and test data.      (b) Training, augmented, and test data.

★ Training data    ● Augmented data    ■ Test data

Figure 6: An example of feature distributions of one few-shot task.

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

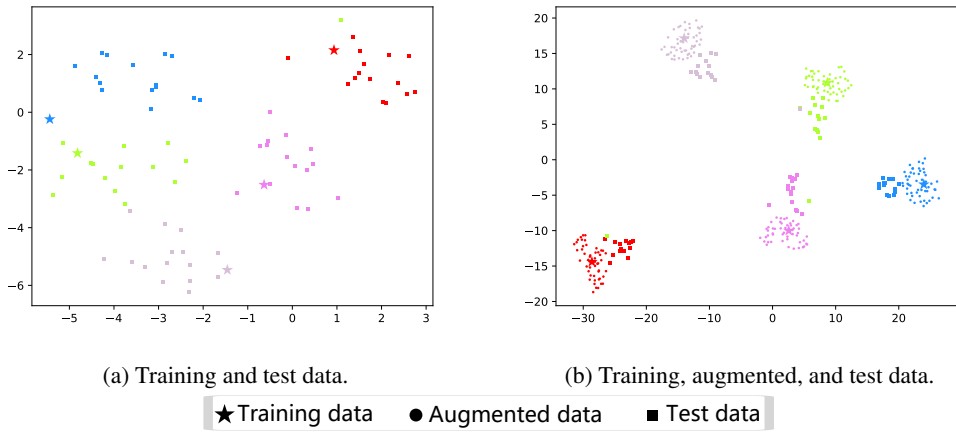

(a) Training and test data.  (b) Training, augmented, and test data.

★ Training data   ● Augmented data   ■ Test data

Figure 11: An example of feature distributions of one few-shot task.

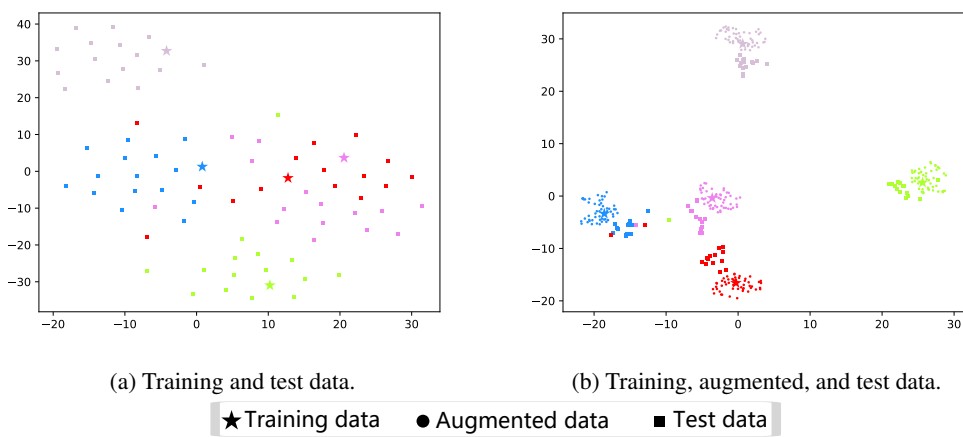

(a) Training and test data.  (b) Training, augmented, and test data.

★ Training data   ● Augmented data   ■ Test data

Figure 12: An example of feature distributions of one few-shot task.

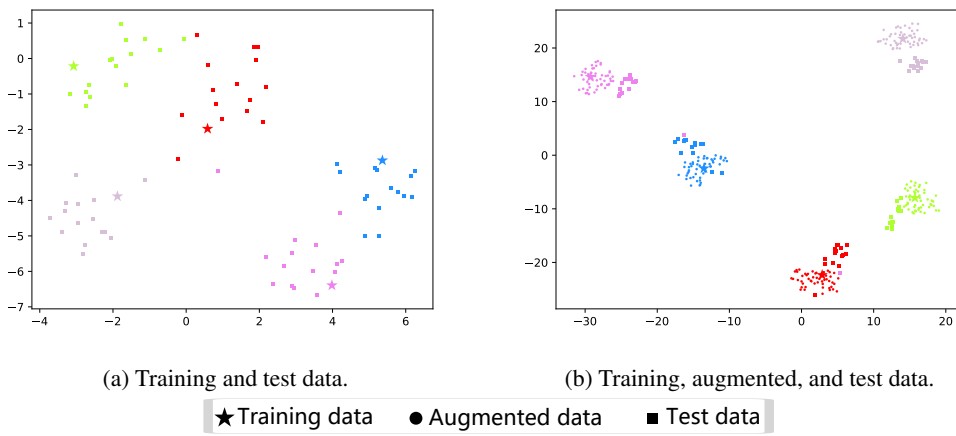

(a) Training and test data.  (b) Training, augmented, and test data.

★ Training data   ● Augmented data   ■ Test data

Figure 13: An example of feature distributions of one few-shot task.

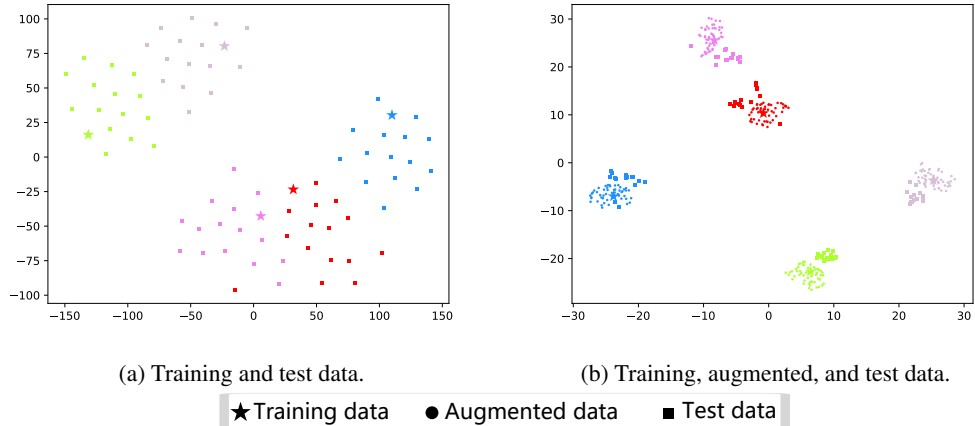

(a) Training and test data.          (b) Training, augmented, and test data.

★ Training data     ● Augmented data     ■ Test data

Figure 14: An example of feature distributions of one few-shot task.

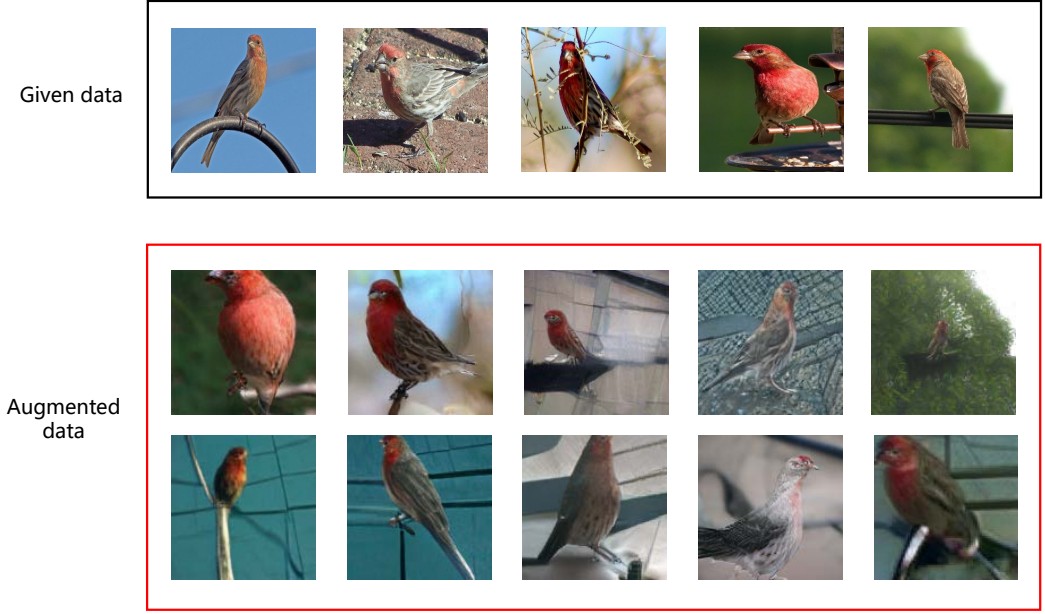

Figure 15: Given data and augmented data in the linnet class.

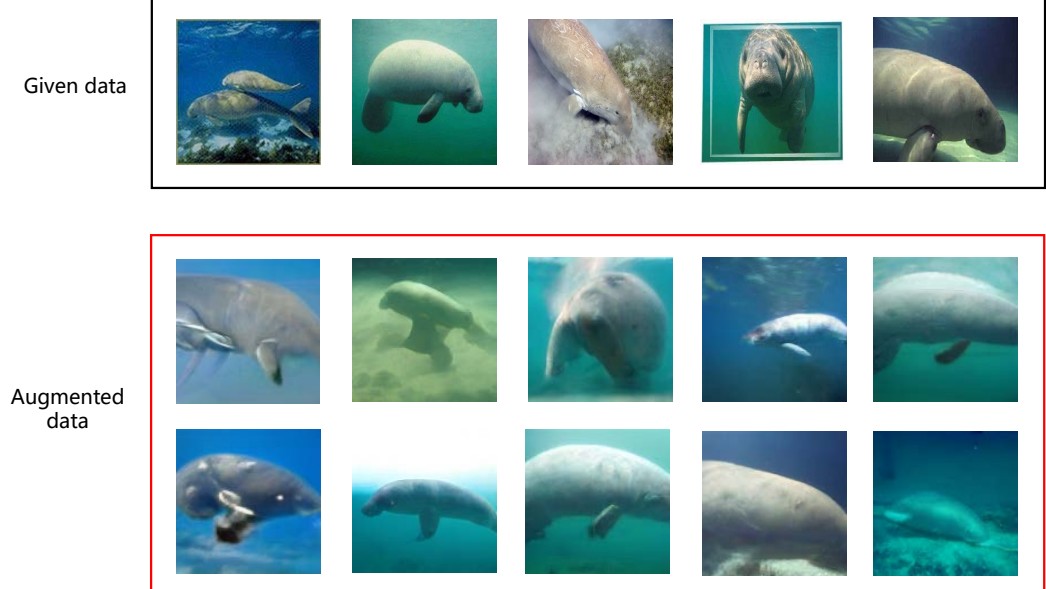

Figure 16: Given data and augmented data in the dugong class.

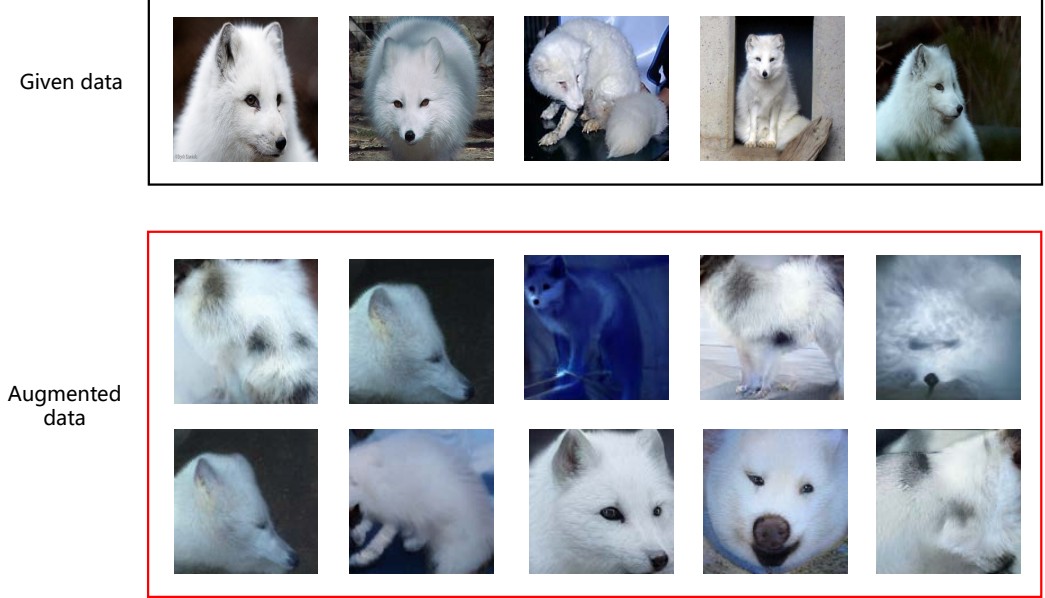

Figure 17: Given data and augmented data in the white fox class.

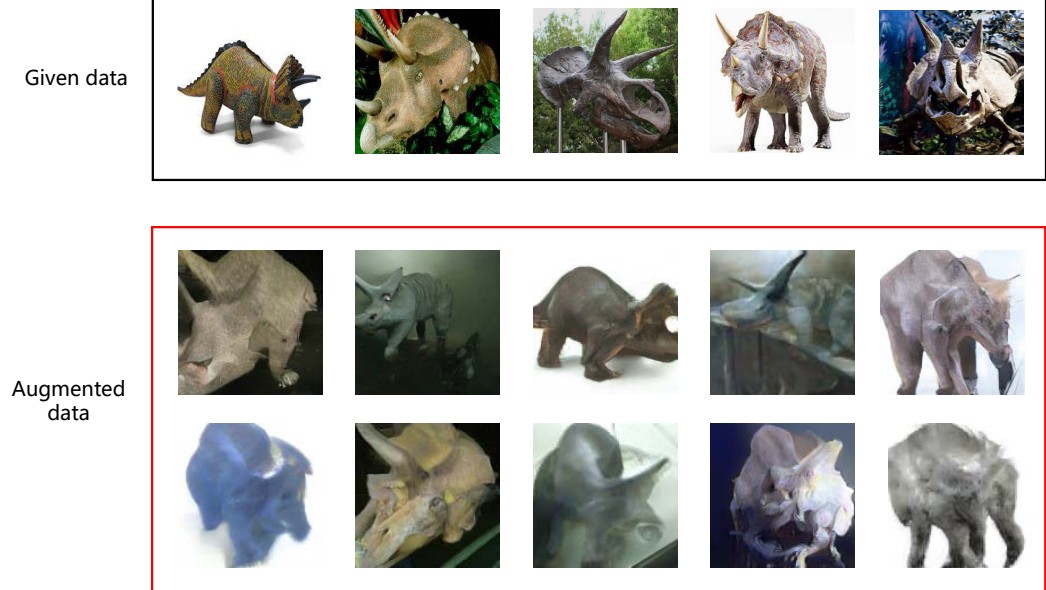

Figure 18: Given data and augmented data in the triceratops class.

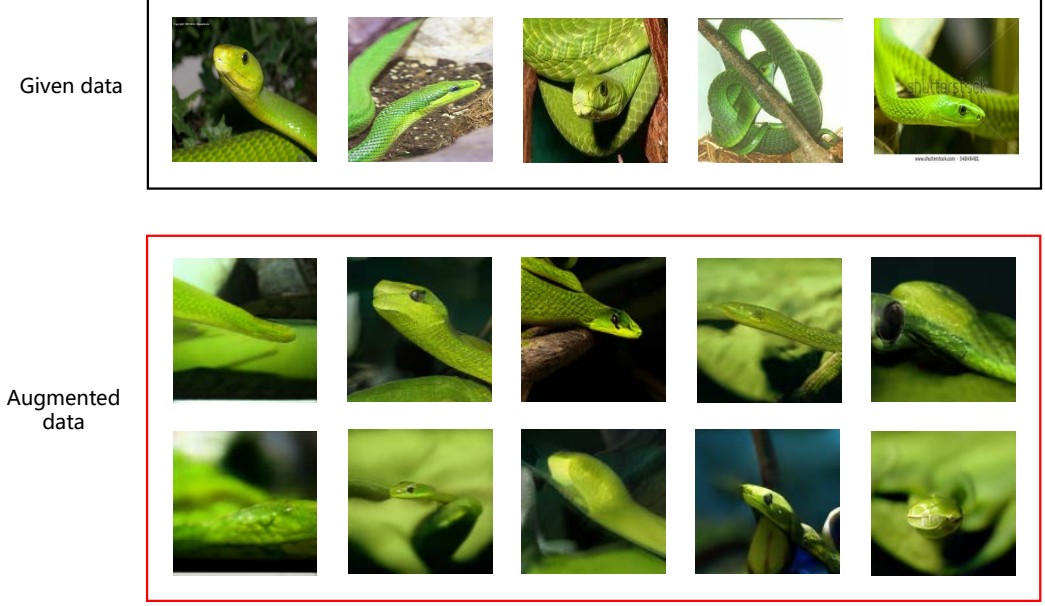

Figure 19: Given data and augmented data in the green mamba class.

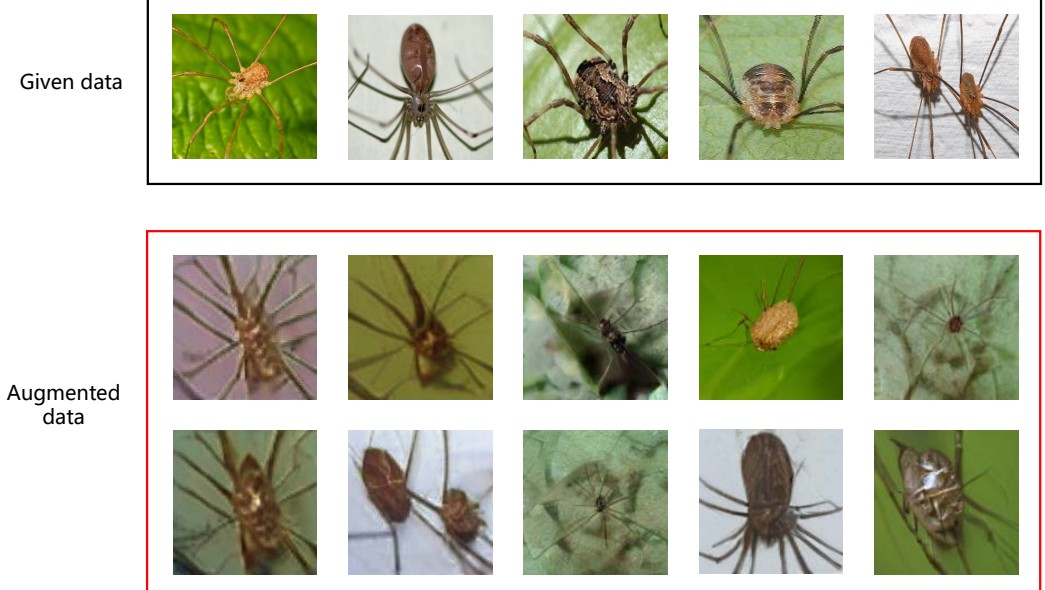

Figure 20: Given data and augmented data in the harvestman class.

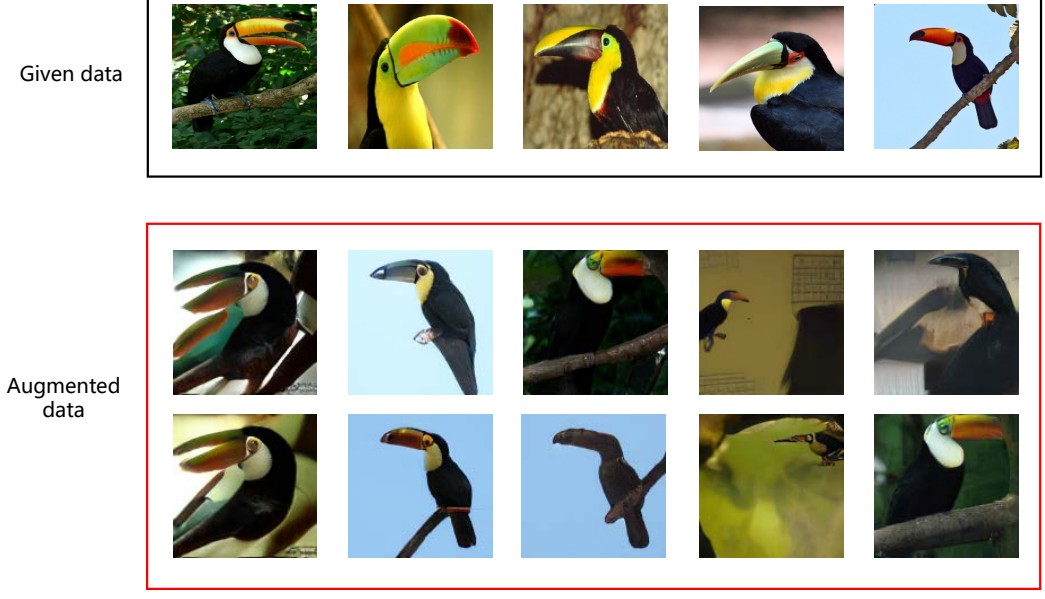

Figure 21: Given data and augmented data in the toucan class.

Given data

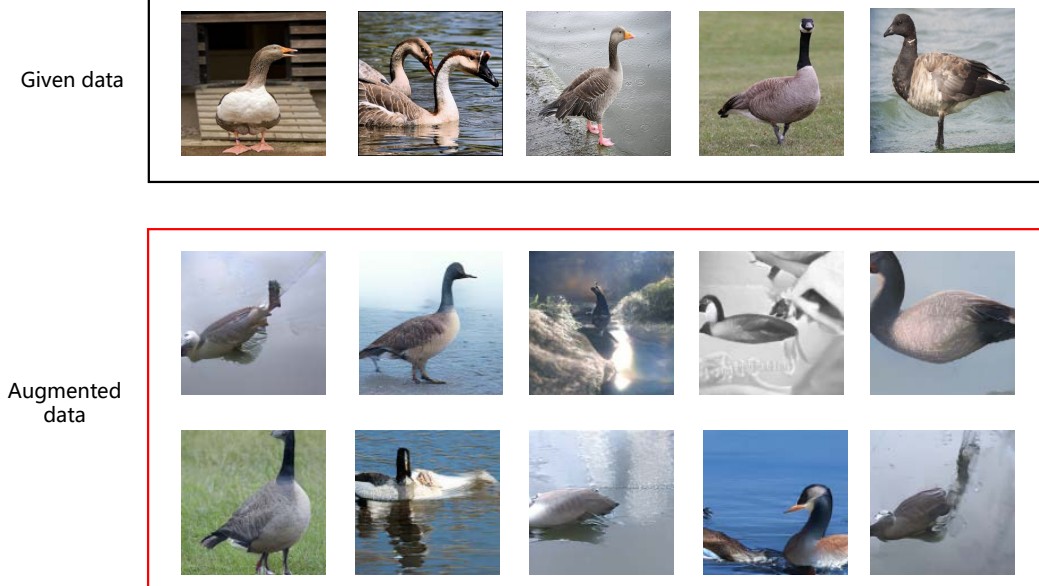

Augmented data

Figure 22: Given data and augmented data in the goose class.

Given data

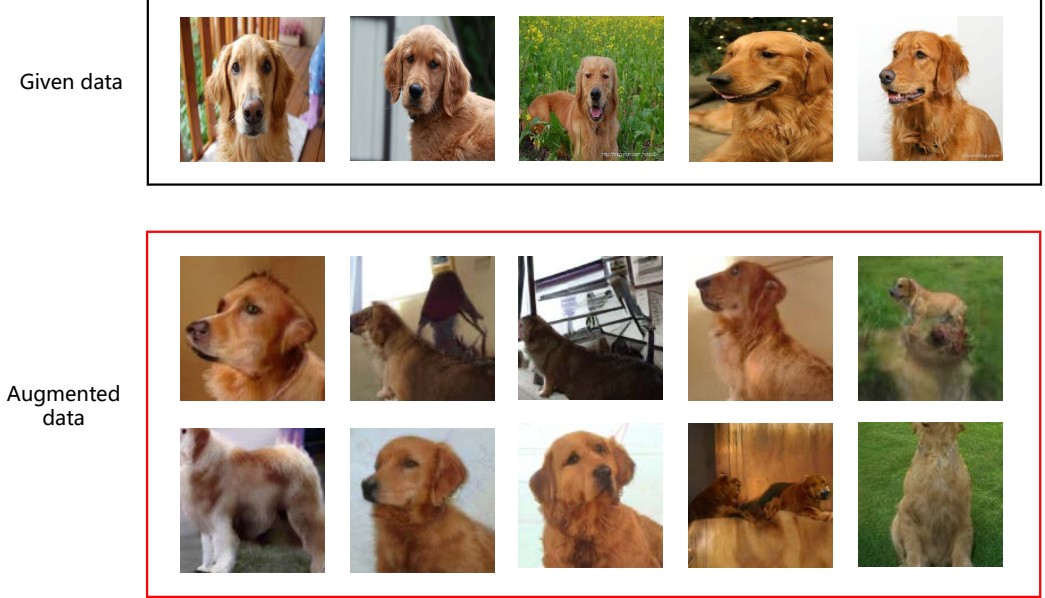

Augmented data

Figure 23: Given data and augmented data in the king crab class.

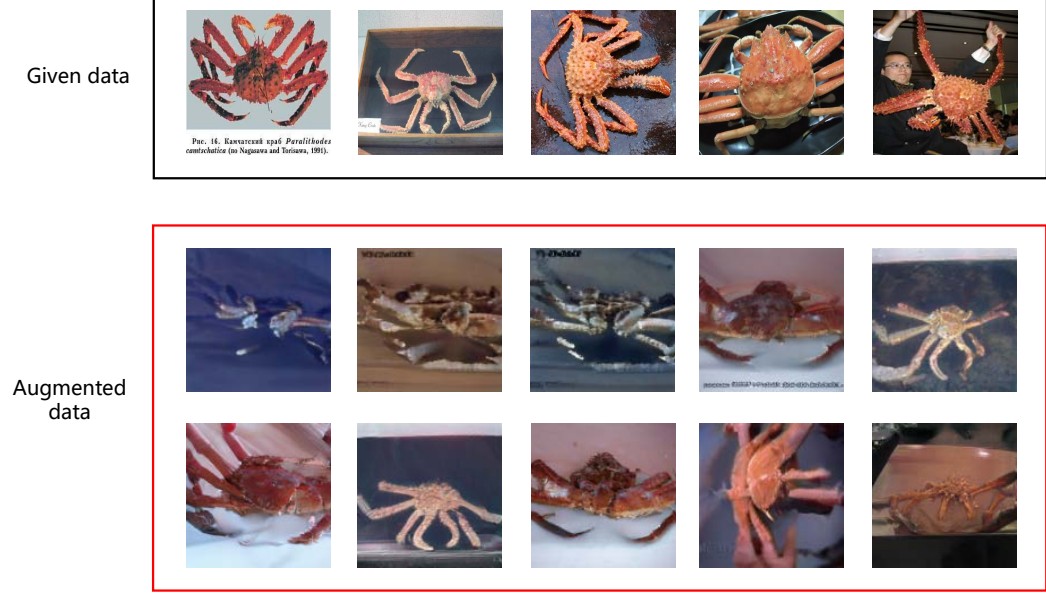

Figure 24: Given data and augmented data in the golden retriever class.