# OpenReview forum: "Hyperbolic Feature Augmentation via Distribution Estimation and Infinite Sampling on Manifolds"
_NeurIPS.cc/2022/Conference — NeurIPS 2022 Accept_

### Official Review · Reviewer_xaq1 · 2022-06-22

**Rating:** 7
**Confidence:** 4
**Soundness:** 3 good
**Presentation:** 3 good
**Contribution:** 4 excellent

**Summary:**

The paper considers the problem of feature augmentation, a special kind of data augmentation, in some low-data regime tasks such as few-shot learning.
The main idea is to consider a wrapped normal distribution for each category and use the reparametrization trick (in the tangent space of the hyperbolic manifold) to obtain a differentiable algorithm for sampling augmented data. A Neural ODE is then used to estimate the parameters of the wrap distribution. Since optimizing the problem for a large number of sampled data can be difficult, the paper proposes in Section 4.4 to upper bound the classification loss and optimize a tractable upper bound in the tangent space, which is Euclidean.
As explained in Section 4.5, the training process is done by exploiting the validation data in a bi-level optimization manner.
Experimental results in the few-shot and continual learning tasks show the relevance of the approach.

**Questions:**

- Have you tried to learn prototypes by using the provided category hierarchy (e.g., on tiered-Imagenet)? It does not seem to be the case.

**Strengths And Weaknesses:**

Strengths: The paper is clear and well written. I think that the approach is original and significant in the hyperbolic manifold learning community. In particular, it proposes different tricks to make the training algorithm tractable so that samples represented on a hyperbolic manifold are high quality (as shown in Figure 3) and improve classification performance.

Weaknesses: The main "weakness" of the approach is that it does not quantitatively outperform baselines by a large margin (many baselines return similar performance on many datasets).
However, I still think that the methodology is interesting and can influence future work in the same direction.

---

> ### Author Response · Authors · 2022-08-02
> **Response to Reviewer xaq1**
>
> Thank you for the positive feedback.
>
> $\textbf{Q1.}$ Have you tried to learn prototypes by using the provided category hierarchy (e.g., on tiered-Imagenet)? It does not seem to be the case.
>
> $\textbf{A1:}$ The prototypes in our work are obtained by averaging features of training data (the prototypes are not learned by using the category hierarchy knowledge). Thank you for the comment. Learning the prototypes in hyperbolic space is an interesting direction. In doing so, we may need to explicitly discover and model the hierarchical relationships of given data, and use such hierarchical relationships to guide representation learning. We will acknowledge your comment as a future direction of study.

---

### Official Review · Reviewer_sUbE · 2022-07-11

**Rating:** 5
**Confidence:** 3
**Soundness:** 3 good
**Presentation:** 3 good
**Contribution:** 3 good

**Summary:**

This paper proposes a hyperbolic feature augmentation method to prevent overfitting when limited data is given. The authors derive an upper bound of the loss function with infinite data augmentation. Therefore, the classifiers and distribution estimators can be trained via a bi-level optimization manner without complex hyperbolic operations. They also provide extensive experimental results on few-shot learning and continual learning to demonstrate the effectiveness of their proposed method.

**Questions:**

see weakness and questions

**Limitations:**

Limitations are not given

**Strengths And Weaknesses:**

Strengths
1.	This paper uses Neural ODEs to estimate the distribution of hyperbolic features and then sampling augmented features from the learned distribution.
2.	This paper provides an upper bound of the augmentation loss using an infinite number of augmentations. This upper bound allows the authors to make efficient augmentation without complex computation.
3.	The authors conduct extensive comparison experiments, ablation experiments and visualization experiments. The results demonstrate the effectiveness of their proposed feature augmentation method.


Weaknesses and Questions
1.	For the distribution estimation, this paper uses three Gradient flow networks to learn different parameters. According to Section 4.3, the networks learn specific parameters for each class using different inputs (i.e., $\bar x_j$). That is to say, the network $F_1$ will output ${c_j}_{j=1}^n$ for $n$ classes. However, in Line 142, the authors point that the parameter $c$ is shared between all classes. How to unify this $c$?
2.	How to update $F_2$, and $F_3$ via minimizing Eq. (16)? When the classifiers are fixed, it seems that only the network $F_1$ can be trained.
3.	Some experimental details are missing.
3.1.	What is the ratio of training data $D_t$ to validation data $D_v$ in the training stage.
3.2.	What is the value of initial $c$.
4.	In Table 2, the metric-based baseline FEAT performs similar accuracy to the proposed method. It’s better to discuss the superiority of the method in terms of time consumption. According to Algorithm 2, the upper bound Eq. (14) simplifies the training of classifiers, but Eq. (16) is still difficult to compute.
Typo: In Table 2, the description does not match the content, e.g., "Euclidean Metric" (or "Hyperbolic Metric") and "Model".

---

> ### Author Response · Authors · 2022-08-02
> **Response to Reviewer sUbE**
>
> Thank you for the feedback. Below, we address your comments individually.
>
> $\textbf{Q1.}$ The network F1 outputs curvatures $\\{c_j\\}_{j=1}^n$ for $n$ classes. How to unify this $c$?
>
> $\textbf{A1:}$ We directly produce a common curvature $c$ for all classes using F1, instead of producing class-specific curvatures $\\{c_j\\}_{j=1}^n$ and unifying them. We send all data of the $n$ classes together to the gradient flow network F1, concatenate all class representations $[\boldsymbol{e}''_1, \boldsymbol{e}''_2, \cdots, \boldsymbol{e}''_n]$, and use the output fully-connect layer in F1 to produce the gradient flow for the common curvature $c$, $\frac{d c^{t}}{ d t} = f_o([\boldsymbol{e}''_1, \boldsymbol{e}''_2, \cdots, \boldsymbol{e}''_n])$. We will clarify our method and add the details in the revised version.
>
>
> $\textbf{Q2.}$ How to update F2, and F3 via minimizing Eq. (16)? When the classifier $\boldsymbol{W}^{\star}$ is fixed, it seems that only the network F1 can be trained.
>
> $\textbf{A2:}$ Eq. (16) uses a bi-level optimization procedure (an inner-loop and an outer-loop) to update F2 and F3. Although the classifier $\boldsymbol{W}^{\star}$ is not changed in the outer-loop, F2 and F3 can still be updated by differentiating through the procedure of the inner loop. Concretely, in the inner-loop, we use F2 and F3 to estimate the distribution parameters $\boldsymbol{\mu}$ and $\boldsymbol{\Sigma}$, and update the classifier $\boldsymbol{W}^{\star}$ by Eq. (14) based on $\boldsymbol{\mu}$ and $\boldsymbol{\Sigma}$. The whole procedure of the inner-loop is differentiable, and the computational graph of the inner-loop is stored in memory.
> In the outer loop, the classifier $\boldsymbol{W}^{\star}$ is fixed, but not a constant. The gradients with respect to F2 and F3 are calculated via backpropagation from $\boldsymbol{W}^{\star}$ to $\boldsymbol{\mu}$ and $\boldsymbol{\Sigma}$, and finally to F2 and F3, according to the stored computational graph. In this case, F2 and F3 are updated.
>
> $\textbf{Q3.}$ What is the ratio of training data $\mathcal{D}_t$ to validation data $\mathcal{D}_v$ in the training stage?
>
> $\textbf{A3:}$ In the few-shot learning experiment, the ratio of $\mathcal{D}_t$ to $\mathcal{D}_v$ follows the standard protocol. In the 1-shot setting, the ratio is $1:15$, that is, we sample one sample as the training data and $15$ samples as the validation data in each task. In the 5-shot setting, the ratio is $1:3$, that is, we sample $5$ sample as the training data and $15$ samples as the validation data in each task. In the continual learning task, the ratio is $1:1$.
>
> $\textbf{Q4.}$ What is the value of initial $c$?
>
> $\textbf{A4:}$ The initial value of the curvature $c$ is set as $-1$.
>
> $\textbf{Q5.}$
> It’s better to discuss the superiority of the method in terms of time consumption. According to Algorithm 2, the upper bound Eq. (14) simplifies the training of classifiers, but Eq. (16) is still difficult to compute.
>
> $\textbf{A5:}$
> Thank you for the comment. We add a wall-clock time experiment and compare our results with the state-of-the-art few-shot learning method FEAT [8]. Concretely, we measure the time consumption (seconds) in the training and test processes. The training time is measured over $10$ epochs, and the test time is measured over $600$ few-shot tasks.
> Results are shown in the following table.
> Experimental results show that our method has comparable time consumption with FEAT, and Eq. (16) is not overly difficult to compute in practice, although it has a seemingly complex optimization procedure. We use a simple hyperbolic gradient descent algorithm and set a small number of iterative steps (only $10$ steps) in the inner-loop, and the involved operations in the hyperbolic gradient descent algorithm are all element-wise operations, which make Eq. (16) not complicated.
>
> Method | 1-shot Training time | 5-shot Training time | 1-shot Test time | 5-shot Test time
> :--: | :--: | :--: | :--: | :--:
> FEAT [8] | $108$ | $151$ | $47$ | $58$
> Ours | $132$ | $195$ | $49$ | $69$
>
> [8] Ye et al. Few-shot learning via embedding adaptation with set-to-set functions. CVPR 2020.
>
> $\textbf{Q6.}$ Typo: In Table 2, the description does not match the content, e.g., "Euclidean Metric" (or "Hyperbolic Metric") and "Model".
>
> $\textbf{A6:}$ Thank you for pointing it out. We will correct them in the revised version.

---

> ### Comment · Reviewer_sUbE · 2022-08-08
> **Comments after the authors' response:**
>
> I've read the authors' responses, and I would like to maintain my recommendation.

---

### Official Review · Reviewer_KDoT · 2022-07-14

**Rating:** 4
**Confidence:** 4
**Soundness:** 3 good
**Presentation:** 3 good
**Contribution:** 2 fair

**Summary:**

This paper introduces hyperbolic feature augmentation. This is done by way of introducing a neural-ODE distribution estimation scheme, whose continuous optimization process can approximate the real data distribution reasonably well in scarce data regimes. An upper bound of the augmentation loss is also derived; the paper makes use of this bound to train the final models without explicit augmentation. Experimental results on several few shot datasets give some evidence for the efficacy of the method.

**Questions:**

### Detailed Comments, Corrections and Questions

Overall the writing in the paper is understandable, but has some grammatical issues.

In terms of questions, I can perhaps reiterate one of the key points from the weaknesses section here: why is the feature augmentation method focused so much on datasets that don't have hyperbolic structure, and moreover, so focused on the extreme data scarce case (few shot) as opposed to more traditional settings? Data augmentation should be helpful in general, not just at the extremes.

A non exhaustive list of minor corrections is given below:

L13: "low data regimes" -> "scarce data regimes". I will only highlight this correction once, but it should be applied many times throughouth the paper.

L58: "in the hyperbolic space" -> "in hyperbolic space"

L305: "much computational loads" -> "much computational load"

**Limitations:**

The paper does reasonably address limitations in the final paragraph. Namely, the paper states the assumption that the data used with their method has a uniform hierarchical structure, whereas in actuality, real-world data may have be more complex and have complex hierarchical structure with varying local structures.

**Strengths And Weaknesses:**

### Strengths

1. The paper is a novel early feature augmentation method in hyperbolic space. Feature augmentation has not been explored much outside of Euclidean space, and I believe this is an interesting area for work.

2. The Table 1 results seem to give reasonable evidence for the efficacy of the introduced method. That is, under the extreme data scarcity seen in the few shot setting, the method does outperform several reasonable baselines, to a statistically significant extent.

### Weaknesses

1. While the method is an early work in hyperbolic feature augmentation, I am unsure as to why the paper focuses so much on few shot learning. All of the test datasets are few shot, and moreover, there is no particular reasons to assume they have an inherently hierarchical structure. This is rather puzzling. If the method is truly a good hyperbolic feature augmentation method, it should be helpful regardless of the few shot setting. I think the experiments are currently quite flawed and incomplete. At the minimum, the paper needs to demonstrate good performance on a reasonably hyperbolic dataset (e.g. the disease dataset from HGCN [b]), and at best, it should do this for several datasets with hyperbolic structure that are outside of the few shot setting.

Certainly, if the margin is the best for the few shot case, so be it. But I would expect cases with actual hyperbolic structure to work better. Moreover, the few shot descriptor should not be an essential test dataset characteristic of a general feature augmentation method.

Ultimately, this additional evaluation will help determine if the improvement is truly due to the geometry captured by the method, or if the improvement is mostly due to some unrelated architectural modification.

2. Regular Euclidean Neural ODEs are used to learn the feature densities, despite the fact that the method is clearly about hyperbolic feature augmentation. This should be augmented by using a Neural Manifold ODE [a] for hyperbolic space.

3. The results in Tables 2 and 4 seem to be relatively marginal. Again this increases my suspicion that the evaluation is not being done on sufficiently hyperbolic datasets to see a real significant difference due to the captured geometry (i.e. due to the fact that the method is hyperbolic and not Euclidean).

### Verdict

Overall this paper has some promise as an early work attempting to do hyperbolic feature augmentation. However, lack of proper evaluation, as described above, make me question the significance of the proposed method. As a result, I recommend a borderline reject rating for this paper.

### References

[a] Neural Manifold ODEs. https://arxiv.org/abs/2006.10254

[b] Hyperbolic GCN. https://arxiv.org/abs/1910.12933

---

> ### Author Response · Authors · 2022-08-02
> **Response to Reviewer KDoT (Part 1)**
>
> Thank you for the review. Below, we address your comments individually.
>
> $\textbf{Q1.}$ The used datasets don't have hyperbolic structures.
>
> $\textbf{A1:}$ We believe that the datasets used in our work have hyperbolic structures. The reasons are as follows.
>
> (1) Some works have shown that the used datasets (i.e., mini-ImageNet, tiered-ImageNet, CUB, and CIFAR) have hyperbolic structures, and the four datasets are commonly used to evaluate hyperbolic algorithms [c,d,e,f,g,h].
> Having an inherent hierarchy is an important factor to make a dataset suitable for hyperbolic representation, and images in these datasets have a lot of inherent hierarchical information.
> For example, the work [f] suggests that the animal images of different species in the CUB dataset have hierarchical structures (e.g., images of birds and parrots have hierarchical structures). The work [c] shows that there are hierarchical structures between an image and its local part images in the mini-ImageNet dataset. We follow these works to use the four datasets.
>
> (2) To further address your concern, we compute the $\delta$-Hyperbolicity [c,i,j] to measure whether the used datasets have hyperbolic structures.
> $\delta$ closer to $0$ indicates a stronger hyperbolic structure of a dataset. The values of $\delta$ are $0.24$, $0.21$, $0.25$, and $0.23$ on the mini-ImageNet, tiered-ImageNet, CUB, and CIFAR-FS datasets, respectively. Please note that some widely used tree-likeness graph datasets have larger $\delta$ than that in the four datasets. For example, $\delta$ of the CORA dataset is $0.35$, and $\delta$ of the PUBMED dataset is $0.36$, where CORA and PUBMED are widely used to evaluate hyperbolic graph neural networks [b].
>
> In addition, we add an extra experiment using the graph data suggested by you and our method comfortably improves upon HGCN [b]. Please see the answer to your question Q3.
>
>
> [b] Chami et al. Hyperbolic Graph Convolutional Neural Networks. NeurIPS 2019.
>
> [c] Khrulkov et al. Hyperbolic Image Embedding. CVPR 2020.
>
> [d] Fang et al. Kernel Methods in Hyperbolic Spaces. ICCV 2021.
>
> [e] Gao et al. Curvature-Adaptive Meta-Learning for Fast Adaptation to Manifold Data. T-PAMI 2022.
>
> [f] Yan et al. Unsupervised Hyperbolic Metric Learning. CVPR 2021.
>
> [g] Guo et al. Clipped Hyperbolic Classifiers Are Super-Hyperbolic Classifiers. CVPR 2022.
>
> [h] Ermolov et al. Hyperbolic Vision Transformers: Combining Improvements in Metric Learning. CVPR 2022.
>
> [i] Adcock et al. Tree-like structure in large social and information. ICDE 2013.
>
> [j] Zhang et al. Lorentzian Graph Convolutional Networks. WWW 2021.
>
>
> $\textbf{Q2.}$ The few shot descriptor should not be an essential test dataset characteristic of a general feature augmentation method.
>
> $\textbf{A2:}$
> The goal of this manuscript is an augmentation method for the scarce data setting, and our method shows the benefit of augmentation when the data is scarce. This case studied in our paper addresses very significant and challenging problems, as compared to data augmentation in the traditional setting. The reasons are as follows.
>
> (1) In the open environment of the real-world, many applications provide scarce data, as collecting and labeling data is high-cost. Data augmentation is a commonly used scheme to solve this problem [k,l].
>
> (2) Compared with the traditional setting with sufficient data, data augmentation in the few-shot setting is more challenging, since capturing the real distribution from scarce data is difficult [m].
>
> Meanwhile, our method can also be applied to the traditional setting. According to your suggestion, we add an experiment on the graph node classification task, and our method leads to improvements (Please see the answer to Q3).
>
>
> [k] Zhang et al. Deep Adversarial Data Augmentation for Extremely Low Data Regimes. T-CSVT 2021.
>
> [l] Li et al. Adversarial Feature Hallucination Networks for Few-Shot Learning. CVPR 2020.
>
> [m] Jiang et al. Deceive D: Adaptive Pseudo Augmentation for GAN Training with Limited Data. NeurIPS 2021.

---

> ### Author Response · Authors · 2022-08-02
> **Response to Reviewer KDoT (Part 2)**
>
> $\textbf{Q3.}$ The paper needs to demonstrate good performance on a reasonably hyperbolic dataset (e.g. the DISEASE dataset from HGCN [b]).
>
> $\textbf{A3:}$ Thank you for the suggestion. We add an experiment on the graph node classification task using the DISEASE, CORA, and PUBMED datasets. Concretely, we use HGCN [b] as the baseline.
> We add our method between the graph convolutional network and the node classifier, and perform augmentation for node features.
> We train the gradient flow networks to estimate data distributions of node features, and use augmented features to train the node classifier.
> We report the average F1-score (%) with the standard deviation on 10 random experiments.
> Results are shown in the following table. This experiment shows that our method leads to improvements in the traditional setting as well.
> For example, on the DISEASE dataset, the F1 score of HGCN is $74.5$%. In contrast, HGCN+Ours achieves $78.0$%, $3.5$% higher than HGCN. We will add the experiment in the revised version.
>
> Method | DISEASE | CORA | PUBMED
> :--: | :--: | :--: | :--:
> HGCN [b] | $74.5 \pm 0.9$ | $79.9 \pm 0.6$ | $80.3 \pm 0.3$
> HGCN+Ours | $\mathbf{78.0 \pm 0.7}$ | $\mathbf{81.1 \pm 0.7}$ | $\mathbf{81.5 \pm 0.2}$
>
> [b] Chami et al. Hyperbolic Graph Convolutional Neural Networks. NeurIPS 2019.
>
>
> $\textbf{Q4.}$ A Neural Manifold ODE [a] should be used for hyperbolic space.
>
> $\textbf{A4:}$ We argue that the wrapped normal distribution should be estimated via the Euclidean neural ODE. Although the wrapped normal distribution is used to model hyperbolic data, the parameters of the distribution are in Euclidean space. The distribution parameters $\boldsymbol{L}$ (decomposed from the augmentation covariance $\boldsymbol{\Sigma}$) and $\boldsymbol{\mu}$ (augmentation mean) are on the tangent space at the origin without the manifold constraint. The curvature $c$ does not have the manifold constraint as well. Thus, we use the Euclidean neural ODE to estimate such parameters in Euclidean space.
>
>
> [a] Lou et al. Neural Manifold Ordinary Differential Equations. NeurIPS 2018.
>
>
>
> $\textbf{Q5.}$ The evaluation is not being done on sufficiently hyperbolic datasets to see a real significant difference in Table 2 and Figure 4.
>
> $\textbf{A5:}$ Our evaluation is conducted on datasets with hyperbolic structures (Please see the answer to Q1 for details), and we argue that the current experimental results in Table 2 and Figure 4 are significant and can show differences between our method and the baselines. The reasons are as follows.
>
>
> (1) We compare our method with existing hyperbolic few-shot learning methods (i.e., CurAML [9] and HyperProto [3]) in Table 2, and our method achieves better performance than them. For example, in the 1-shot setting, our method has more than $3$% improvements compared with the state-of-the-art method CurAML.
>
> (2) We conduct ablation experiments in Table 2 and Figure 4. Compared with `w/o Aug’, our improvements are more than $2$% in the mini-ImageNet dataset, $3$% in the tiered-ImageNet dataset, and $3$% in the CIFAR-100 dataset.
>
> (3) In Table 2 and Figure 4, we also compare our method with some advanced few-shot learning and continual learning methods (e.g, FEAT and MeTAL in Table 2, and IL2A in Figure 4), and the goal is to show that our method can achieve advanced performance as well.
> These advanced methods use some other techniques to improve performance. For example, FEAT uses an extra Transformer model and an auxiliary loss function to refine features, while we directly use the features extracted from the backbone. In this case, our method performs competitively or even exceeds some methods, showing the effectiveness of our method.
>
> [3] Khrulkov et al. Hyperbolic image embeddings. CVPR 2020.
>
> [9] Gao et al. Curvature-adaptive meta-learning for fast adaptation to manifold data. T-PAMI 2022.
>
> [25] Luo et al. Few-shot learning via feature hallucination with variational inference. WACV 2021.
>
> [27] Lazarou et al. Tensor feature hallucination for few-shot learning. WACV 2022.
>
>
>
> $\textbf{Q6.}$ Some grammatical issues.
>
> $\textbf{A6:}$ Thank you for the comment. We will improve the language of this manuscript.

---

> ### Author Response · Authors · 2022-08-09
> **Reply to Reviewer KDoT**
>
> Dear Reviewer KDoT,
>
> We thank you for the review time and valuable comments. We have provided corresponding responses and results, which we believe have covered your concerns. We hope to further discuss with you whether or not your concerns have been addressed. If yes, we would appreciate it if you could improve the rating of our work.
>
> Best,

---

### Official Review · Reviewer_FKNX · 2022-07-26

**Rating:** 8
**Confidence:** 5
**Soundness:** 4 excellent
**Presentation:** 3 good
**Contribution:** 4 excellent

**Summary:**

This paper proposes a hyperbolic feature augmentation method to avoid the overfitting problem of hyperbolic learning in few data setting, the authors extend the method by using an infinite number of augmentations through an interesting upper bound analysis, promising results on few-shot learning and continual learning are given.

**Questions:**

Q1: I think a projection of the sampled noise hat v into the tangent space at the origin is needed?
Q2: can the code, also the reverse mapping algorithm for the visualization made public?
Q3: can the method be made in an end-to-end manner for some example tasks?

**Limitations:**

discussed above.

**Strengths And Weaknesses:**

Strengths:
The motivation of the method is clear and the paper is well-organized/presented, the method is novel/straightforward to understand, the proposal to do learning with infinite data augmentation provides a rigorous and sound way to train the classifier. Experiments on few-shot learning provides promising results in this area.


Weaknesses:
w1. The main weakness is related to the gradient flow network, there are multiple things to model, a practical question is whether these gradient flow networks learn correctly and generalize, especially in a few data setting, it would make the paper stronger if the authors can provide any empirical evaluation of these networks to further verify.

w2: The optimization of the gradient flow networks are a little bit complicated, including both the inner- and outer-loop optimization process + many hyper-parameters, the authors should provide a detailed analysis in this aspect for real usage of this method in practice.

Overall, this paper is novel to me and I'm glad to see hyperbolic models in this setting, the method is clear and straightforward, though a detailed guidance of the method should be made available.

---

> ### Author Response · Authors · 2022-08-02
> **Response to Reviewer FKNX (Part 1)**
>
> Thank you for the positive feedback. Below, we address your comments individually.
>
> $\textbf{Q1.}$ Empirical evaluation of these gradient flow networks to further verify.
>
> $\textbf{A1:}$ Thank you for the suggestion. We use two experiments to demonstrate the generalization ability of the gradient flow network.
>
> (1) In Section 5.2 of the Supplementary Materials, we have conducted a toy experiment by training the gradient flow networks to estimate Gaussian distributions (guided by the MSE loss). We evaluate their performance on unseen Gaussian distributions. Results show that, in the scarce data setting, the trained gradient flow networks can better identify the underlying distribution in comparison to directly computing the distribution parameters from data. For example, when estimating the parameter mean, the MSE error of our method is $2.96 \times 10^4$, far less than the error of directly computing, which is $2.41 \times 10^6$.
>
> (2) We add a new experiment to further demonstrate this point. We randomly generate some hyperbolic wrapped normal distributions, and sample few data from them ($5$ samples from each distribution), where the label of a sample is set by its corresponding distribution. Then we train the gradient flow networks via the bi-level optimization in Eq. (16). Finally, we use the gradient flow networks to recover unseen hyperbolic wrapped normal distributions. Results are that the MSE errors of directly computing distribution parameters from given data are $9.08 \times 10^3$, $5.12 \times 10^5$, and $9.05$ for the mean, covariance matrix, and curvature, respectively. In contrast, the MSE errors of using the trained gradient flow networks are $41.69$, $5.17 \times 10^3$, and $0.42$ for the mean, covariance matrix, and curvature, respectively. These results show that the trained gradient flow networks are capable of generalizing to unseen data in the scarce data setting. We will add the experimental results to the revised version.
>
>
> $\textbf{Q2.}$  A detailed analysis of the inner- and outer-loop optimization process and hyper-parameters for real usage.
>
> $\textbf{A2:}$ In the bi-level optimization process, hyper-parameters include the learning rate in the two loops, the choice of the optimizer in the two loops, and the number of iterative steps in the two loops. The inner-loop optimization plays an important role in efficiently solving the bi-level optimization. An advanced optimizer and a large number of iteration steps in the inner-loop may lead to a good inner-loop solution, benefiting the outer-loop solution. However, this may increase resource consumption and cause the exploding gradient issue. Using a simpler optimizer and a small number of iteration steps can reduce the resource consumption, but it may lead to a biased solution of the inner-loop, resulting in poor performance.
>
> In the implementation, we recommend using a simple gradient descent optimizer in the inner loop to reduce computational complexity, and using the Adam optimizer in the outer-loop to reduce the bias of optimization.
> As for the number of iterative steps, we empirically observe that changing the number of iterations of the inner loop in the range $[5,20]$ does not deteriorate the performance, and setting the number of iterations in the outer-loop larger than $1000$ make the model converge.
> In terms of learning rates of the two loops, we change them in the range $[0.001,0.0001]$, and all get good performance. This shows the robustness of our method to hyper-parameters if they are chosen within a suitable interval.
> We will add the analyses and details to the revised version.
>
>
>
>
> $\textbf{Q3.}$ A projection of the sampled noise $\hat{\boldsymbol{v}}$ into the tangent space at the origin is needed.
>
> $\textbf{A3:}$
> Yes, we indeed have applied such a projection to $\hat{\boldsymbol{v}}$.
> We use the Poincaré ball model of hyperbolic space in this work, and such a projection in the Poincaré ball is defined as a re-scaling function [a,b], i.e., $\hat{\boldsymbol{v}} \gets \frac{\hat{\boldsymbol{v}}}{({\lambda}_{\boldsymbol{p}}^{c})^2}$, where $\boldsymbol{p}$ is the tangent point. In the tangent space at the origin, the projection is $\hat{\boldsymbol{v}} \gets \frac{1}{4} \hat{\boldsymbol{v}}$. We will add it to the revised version.
>
> [a] Nickel et al. Poincaré Embeddings for Learning Hierarchical Representations. NeurIPS 2017.
>
> [b] B´ecigneul et al. Riemannian Adaptive Optimization Methods. ICLR 2019.
>
> $\textbf{Q4.}$ Can the code, also the reverse mapping algorithm for the visualization made public?
>
> $\textbf{A4:}$ Yes, we will release the code of the method and the reverse mapping algorithm, once the manuscript is accepted.

---

> ### Author Response · Authors · 2022-08-02
> **Response to Reviewer FKNX (Part 2)**
>
> $\textbf{Q5.}$ Can the method be made in an end-to-end manner for some example tasks?
>
> $\textbf{A5:}$
> In our implementation, the neural ODE and the feature extractor are trained separately. Meanwhile, our method can be trained in an end-to-end manner as well. Our augmentation process is differentiable, and thus the gradients with respect to the feature extractor can be calculated via backpropagation. We empirically observe that our method achieves similar performance in few-shot learning and continual learning no matter whether it is trained in an end-to-end manner. Thus, we separately train the feature extractor and the neural ODE for less resource consumption and faster training speed (we can directly use the pre-trained feature extractor from other works).

---

### Meta-Review · Area_Chair_PtMC · 2022-08-30

**Recommendation:** Accept
**Confidence:** Less certain

**Metareview:**

This paper attempts to improve learning in hyperbolic space under limited data (few shot setting). In this regards, the authors propose a hyperbolic feature augmentation method to circumvent overfitting. Furthermore, as optimizing using a large number of sampled data can be expensive, the paper proposes an upper bound the classification loss and optimize this tractable upper bound in the tangent space, which is Euclidean making the approach much more practical. There was a wide variance among reviewer scores. We thank the authors and reviewers for actively engaging in discussion and taking steps towards improving the paper including for providing additional experiments. Finally it would be appropriate to tone down the claim that this is the first paper to perform feature augmentation in hyperbolic space as it might be unsubstantiated, cf Weber et al "Robust large-margin learning in hyperbolic space" NeurIPS 2020 which also augments by solving a certification problem.

**Award:**

No

---

### Decision · Program_Chairs · 2022-09-14

Accept